# Wnt/Beta-catenin/Esrrb signalling controls the tissue-scale reorganization and maintenance of the pluripotent lineage during murine embryonic diapause

Rui Fan[1], Yung Su Kim[1], Jie Wu[2], Rui Chen[1], Dagmar Zeuschner[3], Karina Mildner[3], Kenjiro Adachi[4], Guangming Wu[4,5], Styliani Galatidou [1], Jianhua Li[1], Hans R. Schöler [4], Sebastian A. Leidel [2] & Ivan Bedzhov [1✉]

The epiblast, which provides the foundation of the future body, is actively reshaped during early embryogenesis, but the reshaping mechanisms are poorly understood. Here, using a 3D in vitro model of early epiblast development, we identify the canonical Wnt/β-catenin pathway and its central downstream factor Esrrb as the key signalling cascade regulating the tissue-scale organization of the murine pluripotent lineage. Although in vivo the Wnt/β-catenin/Esrrb circuit is dispensable for embryonic development before implantation, autocrine Wnt activity controls the morphogenesis and long-term maintenance of the epiblast when development is put on hold during diapause. During this phase, the progressive changes in the epiblast architecture and Wnt signalling response show that diapause is not a stasis but instead is a dynamic process with underlying mechanisms that can appear redundant during transient embryogenesis.

[1] Embryonic Self-Organization research group, Max Planck Institute for Molecular Biomedicine, Röntgenstraße 20, 48149 Münster, Germany. [2] University of Bern, Department of Chemistry and Biochemistry, Freiestrasse 3, 3012 Bern, Switzerland. [3] Electron Microscopy Unit, Max Planck Institute for Molecular Biomedicine, Röntgenstraße 20, 48149 Münster, Germany. [4] Department for Cell and Developmental Biology, Max Planck Institute for Molecular Biomedicine, Röntgenstraße 20, 48149 Münster, Germany. [5] Guangzhou Regenerative Medicine and Health Guangdong Laboratory, 6 Luoxuan Avenue, Haizhu District, 510320 Guangzhou, China. ✉email: ivan.bedzhov@mpi-muenster.mpg.de

The unbiased developmental capacity of the pre-implantation epiblast to form all somatic lineages, including the germline, is referred to as naive pluripotency[1] and morphologically; the naive epiblast is often described as a simple ball of cells. This ball of pluripotent cells undergoes extensive architectural reorganisation during early embryogenesis to provide the foundation of the future body, but the mechanisms that control the tissue-scale architecture of the epiblast are still poorly understood.

During the pre-implantation stages, pluripotent cells emerge as the result of two subsequent cell fate decisions together with two extra-embryonic tissues: the trophectoderm that later gives rise to the placenta and the primitive endoderm that forms the yolk sac. In mice, by embryonic day four and a half (E4.5), the specification of these three lineages is completed, establishing a mature blastocyst[2]. After the E4.5 embryo hatches from its glycoprotein envelope (Zona pellucida), it comes into direct contact with the mother and initiates implantation into the uterine wall. The attachment of the blastocyst to the uterine surface happens in a restricted period of time known as the implantation window, and this window is determined primarily by the effects of two hormones, ovarian oestrogen and progesterone. Normally, the uterine surface is covered by an anti-adhesive layer of mucins, which functions as a barrier against pathogens, but a temporary spike in the levels of ovarian oestrogen at day 4 of pregnancy leads to the removal of this protective layer, establishing a limited receptive phase that allows the embryo to adhere to the uterine wall. In turn, the cells of the endometrial stroma rapidly proliferate and transform into polyploid decidual cells to support further embryo development by mediating the exchange of nutrients and gases and modulating the entry of immune cells[3,4]. At the same time, the decidua completely engulfs and conceals the implanting embryo, making the peri-implantation development inaccessible for direct studies. Yet, within 24 h, from E4.5 to E5.5, the hollow-shaped blastocyst radically changes its morphology, forming an egg cylinder. During this transition, the naive pluripotent state is dismantled and transforms into formative pluripotency, which has been suggested to capacitate the epiblast for further somatic and germ cell lineage specification[5].

Interestingly, the normal pace of blastocyst development can be put on hold without compromising the pluripotent properties of the epiblast. In an adaptive response to environmental factors that lead to reduced production of ovarian oestrogen, the uterine environment can reside in a non-receptive state, precluding embryo attachment to the uterine wall. Consequently, the embryonic development arrests at the blastocyst stage, and the embryo enters into a reversible state of biosynthetic/metabolic dormancy, known as diapause. In diapause, the transition from naive to formative pluripotency is also halted, maintaining the naive epiblast into a so-called paused state[6,7]. Embryonic diapause can be experimentally induced via ovariectomy or administration of antiestrogenic drugs and sustained for a couple of weeks in mice. The embryo can exit dormancy upon an oestrogen infusion that renders the uterine endometrium in a receptive state, which enables blastocyst attachment and delayed implantation[8].

In previous studies, experimentally inducing diapause was found to increase the efficiency of embryonic stem (ES) cell derivation[9,10]. ES cells are derived from blastocyst-stage embryos and capture the characteristics of the pre-implantation epiblast in vitro. Homogenous ES cell population uniformly expressing naive makers, such as Nanog and Esrrb, can be established and maintained in chemically defined N2B27 medium supplemented with Mek and Gsk3 inhibitors (ground-state culture conditions). Inhibiting Mek suppresses the pro-differentiation Fgf/Erk signals, while inhibiting Gsk3 stabilises β-catenin, boosting the canonical Wnt signalling that relieves Tcf3-mediated transcriptional suppression of pluripotency factors[11,12]. ES cells grown in the presence of these two inhibitors (2i) are transcriptionally and functionally similar to the E4.5 epiblast[13].

Upon withdrawal of 2i from the culture medium, the ES cells downregulate naive makers and exit the pluripotent ground state. In vivo, this transition takes place during the peri-implantation stages, between E4.5 and E5.5, as the epiblast is transformed from a simple ball of cells into a cup-shaped epithelium. This process is characterised by de novo establishment of epithelial polarity and the formation of an apical domain facing a central luminal space (proamniotic cavity). The proper orientation of the apical–basal axis depends on signalling cues provided by the surrounding basement membrane (BM) via integrin receptors expressed on the surface of the epiblast cells[14]. The epithelialization of the pluripotent lineage can be recapitulated in vitro using ES cells grown in a hydrogel of the extracellular matrix (ECM). This three-dimensional (3D) environment mimics the BM niche, enabling similar ES cell self-organisation as the developing E4.5–E5.5 epiblast[14].

As the establishment of epithelial polarity in the epiblast coincides with exit of naive pluripotency during the transition from pre- to post-implantation development, this suggests that the appearance of the pre-implantation epiblast as a ball of non-polarised cells is controlled by pluripotency-associated factors and signalling circuits. Here, we aimed to identify and dissect the signalling cascades and downstream factors that control the morphogenesis of the pluripotent lineage both in vitro—in ground-state ES cells, as well as in vivo—in the pre-implantation epiblast. We find that in vitro, the canonical Wnt/β-catenin signalling negatively regulates ES cells epithelialization via its downstream target—the naive transcription factor Esrrb. In vivo, Wnt signalling is activated in an autocrine manner in the pre-implantation epiblast, controlling the self-organisation and maintenance of the pluripotent lineage during diapause.

## Results

**Active Wnt/β-catenin signalling blocks the establishment of epithelial polarity in a 3D in vitro model of early epiblast development.** The E4.5 epiblast is a ball of pluripotent cells that express naive transcription factors, such as Nanog, that are quickly downregulated as the blastocyst implants and transforms into an egg cylinder. During this transition, the pluripotent lineage undergoes epithelialization, marked by the emergence of Par6-positive apical domain facing the proaminiotic cavity in the centre of the E5.5 epiblast (Fig. 1a). In order to model epiblast morphogenesis in vitro, we utilised a previously established 3D culture method, where ES cells were grown embedded into an ECM hydrogel (Matrigel)[14]. Culturing wild-type E14 ES cells in N2B27 medium for 48 h resulted in the establishment of Par6-positive epithelial rosette-like structures with a central lumen (DMSO control; Fig. 1b, c). At the same time, the naive pluripotent state was dismantled, indicated by the downregulation of Nanog (DMSO control; Fig. 1d). In order to maintain naive pluripotency, we supplemented the medium with 2i, a combination of Mek (PD0325901) and Gsk3 (CHIR99021) inhibitors that sustained Nanog expression while simultaneously blocking the establishment of epithelial polarity (Fig. 1b–d). These observations indicate that the signalling pathways modulated by the small molecules of the 2i cocktail negatively regulate the process of epithelialization.

To discriminate which of the compounds blocks epithelialization, we cultured the cells for 48 h in the presence of DMSO (control), 2i, CHIR99021 (CH) or PD0325901 (PD). Treatment with CH alone was sufficient to block the formation of polarised

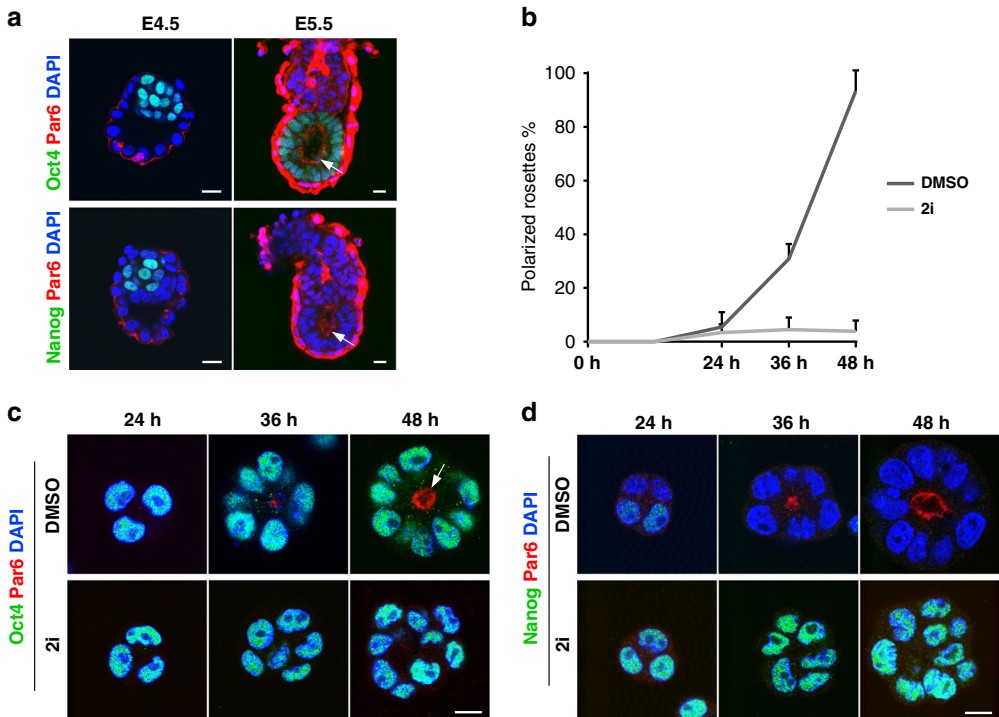

**Fig. 1 ES cells self-organisation in 3D in vitro model of early epiblast morphogenesis. a** E4.5 blastocyst and E5.5 egg cylinder-stage embryos stained for the epithelial polarity marker Par6. The apical domain (arrow) in the E5.5 epiblast surrounds the proamniotic cavity. The epiblast is marked by Oct4 (top panel). The naive pre-implantation epiblast is marked by Nanog (bottom panel), and the nuclei are counterstained with DAPI. **b** Establishment of epithelial rosettes in 3D ES cell culture. Wild-type E14 ES cells were cultured in N2B27 medium supplemented with DMSO (control) or 2i for 24, 36 and 48 h. Data represent mean ± SD, three independent experiments. **c** Wild-type E14 ES cells grown in 3D culture conditions in the presence of DMSO or 2i for 24, 36 and 48 h and stained for Par6, Oct4 and DAPI. The apical domain surrounding the lumen is marked by an arrow. **d** Wild-type E14 ES cells grown in 3D culture conditions in the presence of DMSO or 2i for 24, 36 and 48 h and stained for Par6, Nanog and DAPI. Scale bars, 10 μm.

rosettes, phenocopying the effects of 2i (Fig. 2a, b). As Gsk3 inhibition stimulates the canonical Wnt/β-catenin cascade, this suggested that active Wnt signalling impedes the establishment of epithelial polarity. To validate these results, we stabilised β-catenin genetically using two approaches: by deleting exon-3 of the β-catenin locus that encodes the Gsk3 phosphorylation sites, or via ectopic expression of a Gsk3-mutated β-catenin trans-gene[15], where the Gsk3 phosphorylation sites (Ser-33, Ser-37, Thr-41 and Ser-45) are mutated to alanine (Fig. 2c, d). We used a β-catenin exon-3-floxed ES cell line[16] that expresses a Cre-ERT2 transgene to generate exon-3 excision in the presence of 4-hydroxy tamoxifen (4OHT). This resulted in a block of epithelialization, which recapitulated the effects of the chemical Gsk3 inhibition (Fig. 2c, e and Supplementary Fig. 1a). Similarly, doxycycline-inducible expression of Gsk3-mutated-β-catenin in wild-type E14 ES cells phenocopied the CH treatment of the non-stimulated (-Dox) isogenic control (Fig. 2d, f).

As stabilisation of β-catenin mediated by CH is responsible for impeding epithelialization in 2i culture conditions, we hypothesised that deleting β-catenin should render the cells unresponsive to 2i, enabling the establishment of epithelial polarity. However, β-catenin is not only involved in the canonical Wnt/β-catenin signalling but it is also is a key element of the cell adhesion complex on the cell membrane, bridging E-cadherin and α-catenin (Fig. 2g). Accordingly, conditional knockout of β-catenin via Cre-ERT2-mediated recombination resulted in adhesion defects in both control (DMSO) and 2i-treated cells (Fig. 2h and Supplementary Fig. 1b, c). The β-catenin-lacking ES cells failed to establish a Par6-positive domain and localise apical proteins such as podocalyxin (PDX) on the cell membrane

(Fig. 2h, i), indicating that E-cadherin-mediated intercellular adhesion is essential for proper epithelial integrity.

Therefore, examining the signalling role of β-catenin required an experimental approach compensating for the β-catenin function on the cell membrane. In order to rescue the cell adhesion defect in β-catenin knockout ES cells, we generated a stable β-catenin-floxed Cre-ERT2 ES cell line that expresses an E-cadherin–α-catenin fusion (Eα-fusion) construct (Fig. 2g, j). The Eα-fusion directly links E-cadherin and α-catenin, bypassing the requirement of β-catenin as a bridging factor. As a result, β-catenin ablation upon 4OHT treatment in these cells did not affect the formation of cell–cell contacts resulting in proper self-organisation in control (DMSO) culture conditions. In the presence of 2i, the control β-catenin-expressing cells did not form polarised rosettes, but depletion of β-catenin was sufficient to enable epithelialization (Fig. 2j). Taken together, this shows that the signalling function of β-catenin in the context of the canonical Wnt pathway is responsible for blocking the establishment of epithelial polarity in ground-state culture conditions.

**The Wnt pathway counters epithelialization in the context of naive pluripotency via Esrrb.** Next, we asked whether Wnt/β-catenin signalling blocks epithelialization, specifically in the context of naive pluripotency, and which downstream factors mediate this effect. To address the first question, we compared the self-organisation properties of ES cells in response to Wnt stimulation to more developmentally advanced epiblast-like cells (EpiLC) and epiblast stem cells (EpiSC). The EpiLC are generated from ES cells via treatment with Fgf2 and Activin, which results in the exit of naive pluripotency and a transition into a pre-

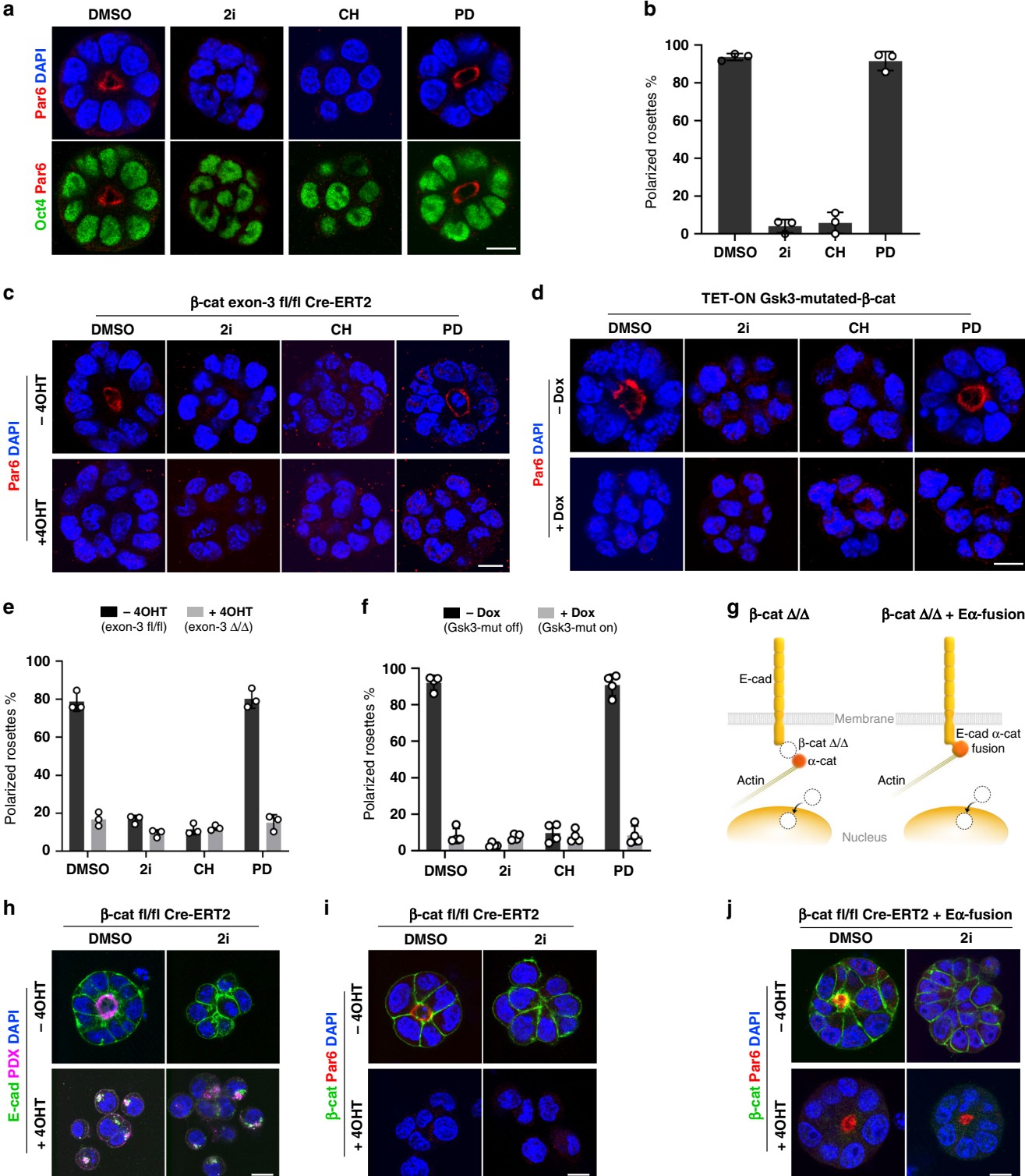

gastrulation epiblast-like state[17]. Conversely, EpiSC are directly derived from early post-implantation embryos and exhibit primed pluripotency[18,19].

In order to directly monitor the Wnt pathway activation, we established ES cells from the TCF/Lef:H2B-GFP reporter mice. This strain harbours a construct containing six copies of a TCF/Lef-responsive element, upstream of a minimal promoter and histone H2B-GFP fusion sequence, enabling faithful capture of Wnt activity at single-cell resolution[20]. As expected, CH treatment of the TCF/Lef:H2B-GFP ES cells resulted in nuclear GFP expression and a block of epithelialization. The Wnt reporter was also activated in TCF/Lef:H2B-GFP EpiLC, but, in contrast to ES cells, the EpiLC remained polarised in the presence of the Gsk3 inhibitor (Fig. 3a). Further analysis at 12, 24 and 48 h of 3D culture showed that EpiLC are polarised epithelial cells and CH treatment in this context induces the expression of early mesendodermal markers, in agreement with the previously reported Wnt function in cell differentiation[21] (Supplementary Fig. 1d, e). Similarly, primed EpiSC also remained epithelial in the CH-supplemented medium (Fig. 3b, c). Thus, active Wnt

**Fig. 2 Active Wnt/β-catenin signalling blocks the establishment of epithelial polarity. a** E14 ES cells grown in 3D culture conditions and stained for Par6, Oct4 and DAPI. **b** Percentage of ES cell clumps that formed Par6-positive polarised rosettes at 48 h of culture. Data represent mean ± SD, three independent experiments. **c** Stabilisation of β-catenin via Cre-mediated exon-3 deletion. β-catenin exon-3 fl/fl Cre-ERT2 ES cells were exposed to 4OHT for 2 days before 3D culture. After that, the cells were culture for 48 h and stained for Par6 and DAPI. **d** Stabilisation of β-catenin via ectopic expression of tetracycline-inducible Gsk3-mutated-β-catenin-IRES-Venus transgene in E14 ES cells. The transgene expression was induced via treatment with Dox for 2 days before 3D culture. The Venus-positive cells were collected by FACS and cultured in Matrigel for 48 h in Dox containing medium supplemented with DMSO, 2i, CH or PD. **e** Percentage of β-catenin exon-3 Δ/Δ and exon-3 fl/fl ES cells that formed Par6-positive rosettes at 48 h of culture. Data represent mean ± SD, three independent experiments. **f** Percentage of control (−Dox) and Dox-treated ES cells harbouring Gsk3-mutated-β-catenin transgene that formed Par6-positive rosettes at 48 h of culture. Data represent mean ± SD, four independent experiments. **g** Schematic representation of the dual role of β-catenin and the experimental approach compensating for the β-catenin function on the cell membrane using E-cadherin-α-catenin fusion. **h** Deletion of β-catenin via Cre-mediated recombination. β-catenin fl/fl Cre-ERT2 ES cells were exposed to 4OHT for 2 days before 3D culture. After that, the cells were cultured for 48 h and stained for E-cadherin, podocalyxin and DAPI. **i** β-catenin-deficient (+4OHT) and control (−4OHT) β-catenin fl/fl Cre-ERT2 ES cells were cultured for 48 h and stained for β-catenin, Par6 and DAPI. **j** Conditional ablation of β-catenin in β-catenin fl/fl Cre-ERT2 ES cells expressing E-cadherin-α-catenin fusion (Eα-fusion). The cells were exposed to 4OHT for 2 days and after that cultured for 48 h and stained for β-catenin, Par6 and DAPI. Scale bars, 10 μm.

signalling counters epithelialization only in ES cells, suggesting that the downstream factors mediating this effect are part of the naive pluripotency network.

To identify Wnt target genes that suppress the establishment of epithelial polarity, we compared the transcriptomes of CH- versus DMSO-treated ES cells grown in 3D culture conditions for 48 h (Fig. 3d, e, Supplementary Fig. 2c and Supplementary Data 1). As an additional reference, we also analysed the transcriptome of cells cultured in the presence of 2i or Fgf2/Activin (Fig. 3d, Supplementary Fig. 2a, b and Supplementary Data 2). Consistent with the establishment of epithelial polarity in the absence of Wnt activation, gene set enrichment analysis (GSEA) showed an increment of focal adhesion, adherens and tight junction expression in DMSO-treated cells (Supplementary Fig. 2d). Next, we probed for naive pluripotency factors upregulated in CH-treated samples in comparison to DMSO. Using available Tcf3 ChIP-seq[22] and Tcf3 knockout RNA-seq data[23], we considered only Tcf3-bound genes, which expression was upregulated upon CH treatment and Tcf3 depletion, as potential candidates. We found 52 genes that met these criteria (Supplementary Fig. 2e and Supplementary Data 3) and we focused on the naive pluripotency factors Klf2, Nr0b1, Tfcp2l1 and Esrrb, including Nanog, (Fig. 3f, g), as they were previously shortlisted as the key pluripotency-associated Wnt targets in mouse ES cells[24]. To determine whether any of the candidates can suppress epithelialization, we generated individual ES cell lines that ectopically expressed each of these factors (Fig. 3h and Supplementary Fig. 3a–c). We found that the Esrrb-expressing cells failed to form polarised rosettes in the DMSO-supplemented medium, phenocopying the CH-mediated block of epithelial polarity (Fig. 3h, i). Accordingly, stabilisation of β-catenin via exon-3 excision was sufficient to sustain Esrrb expression, countering epithelialization in the absence of Gsk3 inhibitor (Supplementary Fig. 3d). In addition, ectopic expression of Esrrb in β-catenin-deficient cells expressing E-cadherin–α-catenin fusion also inhibited the establishment of apical–basal polarity (Supplementary Fig. 3e and Fig. 2j).

Oestrogen-related receptor beta (Esrrb) is an orphan nuclear receptor that is expressed in the naive (non-polarised) epiblast at the blastocyst stage and is downregulated in the post-implantation (polarised) epiblast at E5.5 (Supplementary Fig. 3f). Similarly, in 3D culture, endogenous Esrrb expression was maintained in ground-state culture conditions and shut down in epithelial rosettes formed in the absence of 2i (Supplementary Fig. 3g). Thus, the expression pattern of endogenous Esrrb correlates with the epithelialization of the pluripotent lineage, whereas sustained expression of Esrrb blocks epithelialization, mimicking the effects of active Wnt/β-catenin signalling in naive ES cells.

Since Wnt/β-catenin signalling can suppress epithelial polarity only in naive but not in primed cells (Fig. 3a, b), we asked whether the forced expression of Esrrb would follow the same pattern. To test this, we generated an ES cell line that contains a Dox-inducible Esrrb transgene. These cells were converted to EpiLC via exposure to Fgf2/Activin and then grown in 3D culture for 2 days. Already after 24 h, both control and Dox-treated EpiLC formed polarised rosettes, and they maintained the same number of Par6-positive clusters at 48 h of culture (Fig. 3j, l). This suggests that Esrrb can suppress epithelialization only in the context of naive pluripotency, similar to the effects of CH treatment in ES cells and EpiLC.

To further examine this concept, we converted EpiLC back to ES cells and asked whether Esrrb can also inhibit established polarity during the reprogramming to naive pluripotency. As it was recently shown that in the presence of Lif, ectopic Esrrb expression could convert EpiSC to ES cells[25], we followed the same approach and added Lif to the Dox-supplemented medium. This resulted in the upregulation of endogenous Nanog expression within the first 24 h of 3D culture, revealing an ongoing (re)establishment of naive pluripotency (Fig. 3k). Only after this, between 24 and 48 h of reprogramming, the cells started to lose their epithelial phenotype (Fig. 3j, l). This suggests that reinstating the naive state is a prerequisite for dismantling already established epithelial polarity in more developmentally advanced pluripotent cells.

**Esrrb controls the epithelial programme via Spry2.** As our results indicate that Esrrb expression blocks epithelialization downstream of Wnt/β-catenin, we supposed that deleting Esrrb should render the cells unresponsive to Wnt stimulation, enabling the establishment of epithelial polarity in CH-supplemented medium. To test this, we used Esrrb-floxed ES cell line that expresses tamoxifen-inducible Cre recombinase (MerCreMer)[25] for ablating Esrrb upon 4OHT treatment (Fig. 4a). In the presence of CH, the control Esrrb floxed cells remained non-polarised in 3D culture conditions. In contrast, the deletion of Esrrb resulted in the formation of polarised rosettes, despite the Gsk3 inhibition (Fig. 4b, c). Esrrb knockout cells exhibited epithelialization already at 24 h of the culture, with an increasing rate of apoptosis during the last day of culture revealed by cleaved Caspase-3 staining (Fig. 4b). This shows that in addition to its function as a central Wnt node controlling the establishment of apical–basal polarity, Esrrb is also required for ES cell maintenance.

To further examine the loss of function of Esrrb in the context of active Wnt signalling, we analysed the transcriptional profile of Esrrb null and Esrrb floxed cells grown in 3D culture for 24 h in the presence of CH. Loss of Esrrb resulted in significant

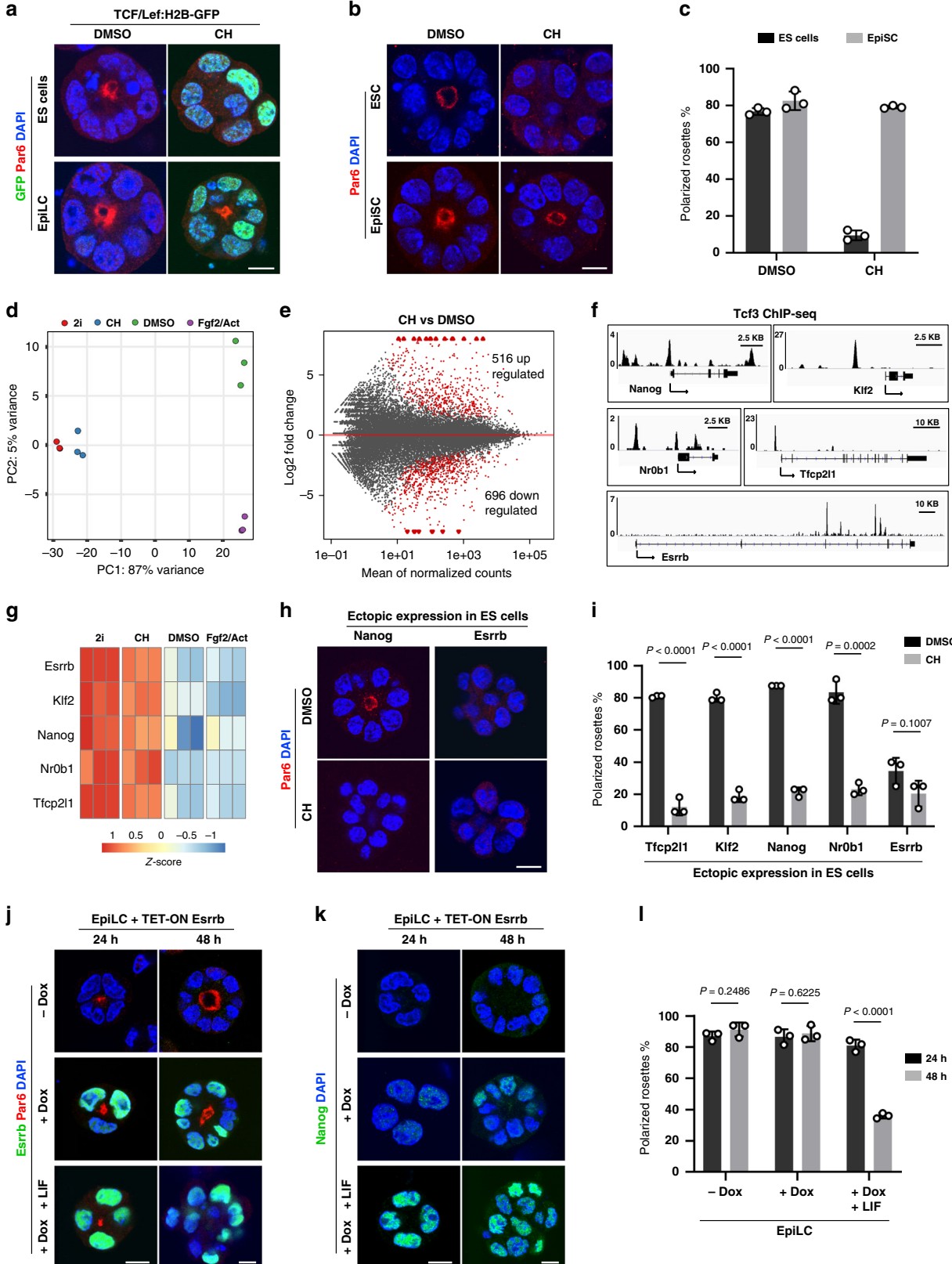

upregulation of 605 genes and downregulation of 128 genes, including naive pluripotency factors, such as Nr0b1 and Klf2 (Fig. 4d, e, Supplementary Fig. 4a and Supplementary Data 4). Gene set enrichment analysis (GSEA) showed an increment of pro-differentiation MAPK signalling and apoptotic genes in Esrrb-deficient cells, consistent with the role of Esrrb in ES cell

self-renewal. In addition, focal adhesion and intercellular tight junction gene sets were enriched in Esrrb knockout cells, indicating ongoing epithelialization (Fig. 4f and Supplementary Fig. 4b).

In order to identify the downstream effectors of Esrrb that control the establishment of apical–basal polarity in a Wnt-

**Fig. 3 Wnt signalling counters epithelialization in the context of naïve pluripotency via Esrrb. a** TCF/Lef:H2B-GFP ES cells and TCF/Lef:H2B-GFP EpiLC stained for GFP, Par6 and DAPI. **b** ES cells and EpiSC cultured in the presence of DMSO or CH and stained for Par6 and DAPI. **c** Percentage of ES cells and EpiSC that formed Par6-positive polarised rosettes. Data represent mean ± SD, three independent experiments. **d** Principal component analysis (PCA) plot of RNA-seq datasets. The cells were grown in 3D culture for 48 h, each culture condition is represented by three replicates. **e** Scatter plot of gene expression of CH and DMSO-treated cells with differentially expressed genes that are significantly up- or downregulated shown in red, adjusted P value < 0.01, three replicates per culture condition. **f** Gene tracks representing the binding of Tcf3 at the indicated loci. The x axis represents the linear sequence of genomic DNA, and the y axis represents the total number of mapped reads. **g** Expression of Wnt target genes with respect to the mean expression across DMSO, 2i, CH or Fgf2/Activin culture conditions. **h** E14 ES cells expressing ectopically Nanog or Esrrb transgenes, cultured in the presence of DMSO or CH and stained for Par6 and DAPI. **i** Percentage of ES cells ectopically expressing Tfcp2l1, Klf2, Nanog, Nr0b1 or Esrrb that formed Par6-positive polarised rosettes. Data represent mean ± SD, three independent experiments, two-tailed unpaired Student's t test, the exact P value is noted in the figure. **j** EpiLC expressing inducible Esrrb transgene were cultured without Dox (control), in the presence of Dox or in medium supplemented with both of Dox and Lif. After 24 and 48 h, the cells were stained for Par6, Esrrb and DAPI. **k** Endogenous Nanog expression during EpiLC reprogramming. **l** Percentage of Par6-positive polarised rosettes at 24 and 48 h of EpiLC reprogramming. Data represent mean ± SD, three independent experiments, two-tailed unpaired Student's t test, the exact P value is noted in the figure. Scale bars, 10 μm.

dependent manner, we examined the genes differentially expressed upon Esrrb depletion and intersected this list with genes which modulate their expression in CH vs DMSO-treated samples. Using available Esrrb ChIP-seq data[26], we considered only Esrrb-bound genes (Supplementary Fig. 4c and Supplementary Data 5) and picked out candidates with a previously assigned regulative or structural role in epithelial tissues. We shortlisted four genes—Arl4c, Krt18, Ntn1 and Spry2 (Fig. 4g). Arl4c (ADP-ribosylation factor-like 4c) is a small GTPase that acts as a molecular switch promoting the activation of Cdc42, which is a central factor orchestrating apical–basal polarity in epithelial cells[27]. Krt18 (Keratin 18) forms intermediate filaments that are essential cytoskeletal components of polarised epithelia[28]. Ntn1 (Netrin-1) controls Par complex localisation during axon guidance[29], and Spry2 (Sprouty2) is a modulator of tyrosine receptor kinase signalling that was previously shown to repress the polarised epithelial phenotype of cancer cells[30]. These candidates were also differentially expressed in wild-type ES cells; Arl4c and Krt18 were upregulated in polarised rosettes (DMSO and Fgf2/Activin culture conditions), whereas Ntn1 and Spry2 expression was associated with the non-polarised phenotype of ES cells grown in CH- and 2i-supplemented media (Supplementary Fig. 4d). In addition, further analysis of Esrrb genome occupancy using the available ChIP-seq data[26] revealed that these candidate genes are bound by Esrrb, indicating that they are direct downstream targets (Fig. 4h).

To understand whether any of the candidates can modulate (promote or oppose) the establishment of apical–basal polarity, we generated stable ES cell lines that ectopically express each of these factors (Supplementary Fig. 4e). These cells were cultured for 48 h in the presence of CH or DMSO and, after that, stained for polarity (Par6) and naïve pluripotency (Nanog) markers. We found that in a CH-supplemented medium, none of the factors were able to induce epithelialization. However, in DMSO culture conditions, sustained Spry2 expression significantly reduced the number of Par6-positive ES cell clumps (Fig. 4i, j and Supplementary Fig. 4f). The Spry2-expressing cells were Nanog-negative and E-cadherin-positive, indicating that the loss of apical domain identity is not due to a delayed exit of naïve pluripotency or cell adhesion defect (Fig. 4j, k). Still, a large portion of the ES cell clumps established apical–basal polarity, suggesting that additional factors are involved in regulating the epithelial programme downstream of Esrrb.

**Establishment of epithelial polarity in the epiblast follows the dynamics of Wnt signalling activity during diapause.** As our in vitro analysis showed that Wnt/β-catenin/Esrrb signalling regulates the self-organisation and maintenance of naïve ES cells, we next asked whether this pathway plays a similar role in vivo, in

the pre-implantation epiblast. To directly monitor Wnt signalling activity, we derived E4.5, E5.5 and E6.5 embryos from the TCF/Lef:H2B-GFP mouse line[20]. The E4.5 blastocysts displayed no Wnt activity above the detection level of the reporter, except for a few cells in the epiblast (Fig. 5a, d). After implantation, the visceral endoderm cells of the E5.5 egg cylinder were GFP-positive, whereas at E6.5, the epiblast cells that displayed reporter activity were located in the posterior region (Fig. 5a). Although we detected only a small amount of Wnt reporter activity in the naive epiblast, none of the previously described Wnt pathway loss-of-function mutants were associated with developmental defects in the pluripotent lineage before implantation[31]. β-catenin knockout embryos show anterior–posterior patterning defects at the egg cylinder stage[32,33], whereas Esrrb is required much later, at E10.5, for proper placentation[34]. Thus, at first look, there is a discrepancy between the role of Wnt/β-catenin/Esrrb signalling in ES cells and during early embryonic development. Notably, in the embryo the naive epiblast exists only transiently at E4.5, whereas in ES cells the naive state is sustained over the long-term, potentially indefinitely. Yet, in vivo conditions also allow for an extended time period where the pre-implantation epiblast is maintained during embryonic diapause. Accordingly, our analysis of diapause blastocysts at equivalent days of gestation (EDG) 5.5, 7.5 and 9.5 showed sustained Nanog expression similar to E4.5 control embryos (Fig. 5b). Moreover, a previously published transcriptional profile of epiblast cells at EDG6.5[35] revealed maintained expression of other naïve markers such as Esrrb, Klf2, Klf4, Nr0b1, Tfcp2l etc. (Supplementary Fig. 5a). Therefore, we asked whether the Wnt pathway is activated in the pluripotent lineage during diapause and whether β-catenin and Esrrb play a role in this process.

To address the first question, we derived EDG5.5, 7.5 and 9.5 diapause embryos as well as E4.5 blastocysts from the TCF/Lef:H2B-GFP reporter strain. We found an increasing Wnt reporter response in the epiblast at EDG5.5, peaking at EDG7.5, followed by a decrease of reporter activity by EDG9.5 (Fig. 5c, d). These observations were in accord with the available transcriptional data of the EDG6.5 epiblast[35], where Wnt target genes, such as Axin2, Cd44, Cdx1, Pitx2, Sp5, as well as Tcf4 (Tcf7l2) and Wnt4 ligand, were upregulated in comparison to E4.5 (pre-implantation) and E5.5 (post-implantation) epiblasts (Supplementary Fig. 5b). Moreover, the decrease of Wnt activity after EDG7.5 was associated with the establishment of epithelial polarity, and by EDG9.5 the majority of the diapause blastocysts showed Par6 expression in the centre of the epiblast (Fig. 5e, f). To further examine the self-organisation of the pluripotent lineage, we performed electron microscopy analysis of E4.5 and diapause embryos. At E4.5, the epiblast appeared as a coherent ball of cells, whereas at EDG9.5 the cells were reorganised into a rosette-like

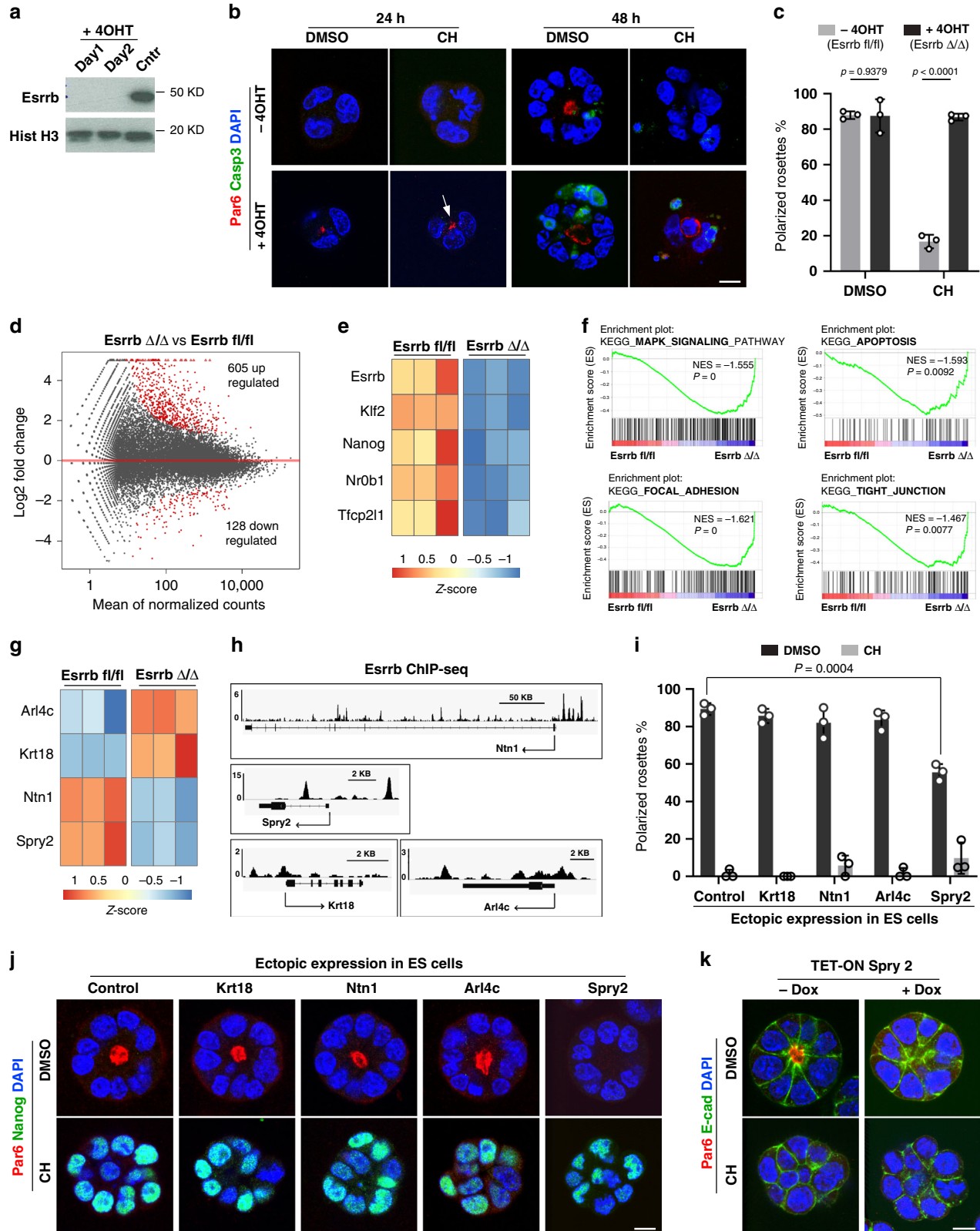

structure. The apical membrane domain surrounded the central microlumen, confirming the establishment of the epithelial phenotype in the epiblast (Fig. 5g and Supplementary Fig. 6).

Next, we asked whether β-catenin plays a role in the epiblast during diapause. We analysed embryos from heterozygous intercrosses at EDG5.5 and found that β-catenin-deficient blastocysts exhibited a complete loss of pluripotent cells, whereas

the extra-embryonic lineages were still present. At the same time, the blastocoel cavity of the knockout embryos appeared collapsed, indicating a potential failure in trophectoderm integrity that may have been the result of inadequate intracellular adhesion (Fig. 6a–c). Such a structural defect was not observed in Esrrb knockout embryos, which retained their blastocyst architecture during diapause. At EDG5.5, the Esrrb-deficient blastocysts were

**Fig. 4 Esrrb controls the epithelial programme via Spry2. a** Depletion of Esrrb upon 4OHT treatment. Histone H3 serves as a loading control. Uncropped blots are provided in source data file. **b** Esrrb fl/fl ES cells expressing tamoxifen-inducible Cre cultured without or in the presence of 4OHT for 1 day before 3D culture. After that, the cells were grown in Matrigel in a medium supplemented with DMSO or CH for 24 h or 48 h and stained for Par6, cleaved Caspase-3 and DAPI. The apical domain in Esrrb-deficient cells is indicated with an arrow. **c** Establishment of epithelial rosettes in Esrrb Δ/Δ (+4OHT) and control Esrrb fl/fl (−4OHT) ES cells, Data represent mean ± SD, three independent experiments, two-tailed unpaired Student's $t$ test, the exact $P$ value is noted in the figure. **d** Scatter plot of gene expression of Esrrb Δ/Δ and Esrrb fl/fl cells grown in the presence of CH for 24 h. Differentially expressed genes that are significantly up- or downregulated are shown in red, adjusted $P$ value < 0.01, three replicates per genotype. **e** Gene expression of Wnt-responsive naive pluripotency factors in Esrrb Δ/Δ and Esrrb fl/fl cells. **f** GSEA plots showing enrichment of MAPK, apoptosis, focal adhesion and tight junction KEGG pathways. **g** Gene expression of epithelial factors in Esrrb Δ/Δ and Esrrb fl/fl cells. **h** Gene tracks representing the binding of Esrrb at the indicated loci. The $x$ axis represents the linear sequence of genomic DNA, and the $y$ axis represents the total number of mapped reads. **i** Percentage of E14 ES cells ectopically expressing Krt18, Ntn1, Arl4c or Spry2 that formed Par6-positive polarised rosettes. Data represent mean ± SD, three independent experiments, two-tailed unpaired Student's $t$ test, the exact $P$ value is noted in the figure. **j** E14 ES cells (control) and E14 ES cells ectopically expressing Krt18, Ntn1, Arl4c or Spry2 stained for Par6, Nanog and DAPI. **k** Inducible ectopic expression of Spry2 in E14 ES cells (+Dox), compared to unstimulated (−Dox) control, stained for Par6, E-cadherin and DAPI. Scale bars, 10 μm.

---

indistinguishable from the heterozygous and wild-type littermates (Supplementary Fig. 7a, b). However, a significant reduction of the number of epiblast cells was detectable in Esrrb knockout embryos at EDG7.5 and EDG9.5 (Figs. 6d–g), suggesting that Esrrb is involved in the maintenance of the pluripotent lineage during diapause.

To understand whether Esrrb expression during diapause is controlled by Wnt/β-catenin signalling, we analysed Esrrb expression at EDG7.5 and EDG9.5 and found that Esrrb levels correlate with the Wnt signalling dynamics—high at EDG7.5 and low at EDG9.5 (Supplementary Fig. 7c, d). We hypothesised that maintaining active Wnt signalling will sustain Esrrb expression at EDG9.5 and counter epithelialization. To test this, first, we aggregated ES cells harbouring β-catenin exon-3 deletion or control (exon-3 floxed) cells with E2.5 morulae expressing Histone H2B:EGFP reporter allele[36]. Thus, we generated chimeric blastocysts comprised of wild-type (EGFP+) extra-embryonic tissues and EGFP-epiblast derived from the donor cells. The embryos were then transferred into foster mothers, where diapause was induced via ovariectomy and subsequently recovered at EDG9.5 for analysis (Fig. 7a, b). We found that blastocysts expressing the stabilised form of β-catenin displayed higher levels of Esrrb and suppressed epiblast polarisation compared to control embryos (Fig. 7b–d).

Second, we asked whether sustaining the expression of Esrrb alone, without experimentally modulating the Wnt pathway, is sufficient to block the establishment of epithelial polarity at EDG9.5. Using the same strategy, we aggregated ES cells that constitutively express Esrrb transgene or control wild-type ES cells with host EGFP+ morulae to generate chimeric blastocysts. After embryo transfer and induction of diapause, the embryos were isolated at EDG9.5. Control embryos exhibited epiblast epithelialization, whereas the pluripotent lineage of embryos ectopically expressing Esrrb remained non-polarised (Fig. 7e, f).

Taken together, this shows that the canonical Wnt/β-catenin signalling controls Esrrb expression in the epiblast and the Wnt/β-catenin/Esrrb cascade counters epithelialization and maintains the pluripotent lineage during embryonic diapause.

**Wnt signalling is activated in an autocrine manner in the diapause embryo.** Finally, we investigated the origin of ligands that drive the activation of the Wnt pathway during diapause. The two potential sources are the embryo itself (autocrine stimulation) or the uterine tissues (paracrine stimulation). To distinguish between these two options, we crossed TCF/Lef:H2B-GFP reporter mice and Wntless (Wls) heterozygous animals to generate a Wls +/− TCF/Lef:H2B-GFP strain. Wls is a trans-membrane protein localised in Golgi, secretory vesicles and the cell membrane and is required for the secretion of Wnt ligands[37].

As Wls null embryos are not able to release the ligands, activation of the TCF/Lef:H2B-GFP reporter in these embryos would indicate an external source of Wnt; and vice versa: if the TCF/Lef:H2B-GFP reporter is not activated in Wls-deficient blastocysts, then Wnt stimulation depends on endogenous Wnt ligands produced by the embryos.

At EDG7.5, the control Wls (+/− and +/+) TCF/Lef:H2B-GFP embryos showed reporter activity in the epiblast. However, the TCF/Lef:H2B-GFP reporter was inactive in Wls-deficient blastocysts, suggesting that endogenously produced Wnt ligands stimulate the Wnt pathway in an autocrine manner (Fig. 8a, b). In addition, Esrrb expression was diminished in Wls knockout mutants at EDG7.5, (Fig. 8c, d), supporting the notion that Wnt signalling controls Esrrb expression during diapause. Although at EDG5.5 the number of epiblast cells in Wls knockout blastocysts was similar to that of the controls (Supplementary Fig. 7e, f), at EDG7.5 we observed a reduction in the number of epiblast cells and epithelialization of the pluripotent lineage in Wls-deficient embryos (Fig. 8e, f), confirming the role of Wnt signalling in the long-term maintenance and self-organisation of the epiblast during diapause.

## Discussion

All tissues of the foetus originate from a small ball of naive pluripotent cells located inside the blastocyst. In contrast to the trophectoderm and primitive endoderm, which establish apical–basal polarity during the pre-implantation stages, the epiblast remains non-polarised until E4.5[14]. These features are captured in vitro by ES cells grown in a 2i-supplemented medium, suggesting that the signalling pathways modulated in these culture conditions control the tissue morphology of the pluripotent lineage. Indeed, we found that Gsk3-inhibitor-mediated activation of the canonical Wnt/β-catenin pathway blocks the establishment of epithelial polarity (Fig. 8g). The β-catenin function in the cell adhesion complex is required for tissue integrity, whereas its nuclear function is critical for transmitting the signalling cues that inhibit the epithelial phenotype. The latter is a context-dependent effect that only ES cells exhibit; more developmentally advanced EpiLCs and EpiSCs do not exhibit this effect, indicating an essential role of downstream naive pluripotency factors.

We found that the naive transcription factor Esrrb is a central Wnt target that counters epithelialization, in addition to its previously described role in ES cells' self-renewal and EpiSC reprogramming to naive pluripotency[24,25,38]. We identified Spry2 as one of potentially a group of effectors that Esrrb regulates, supressing the establishment of apical–basal polarity. It was previously shown that Spry2 plays a tumorigenic role in colon cancer via dysregulating tight junction and epithelial polarity

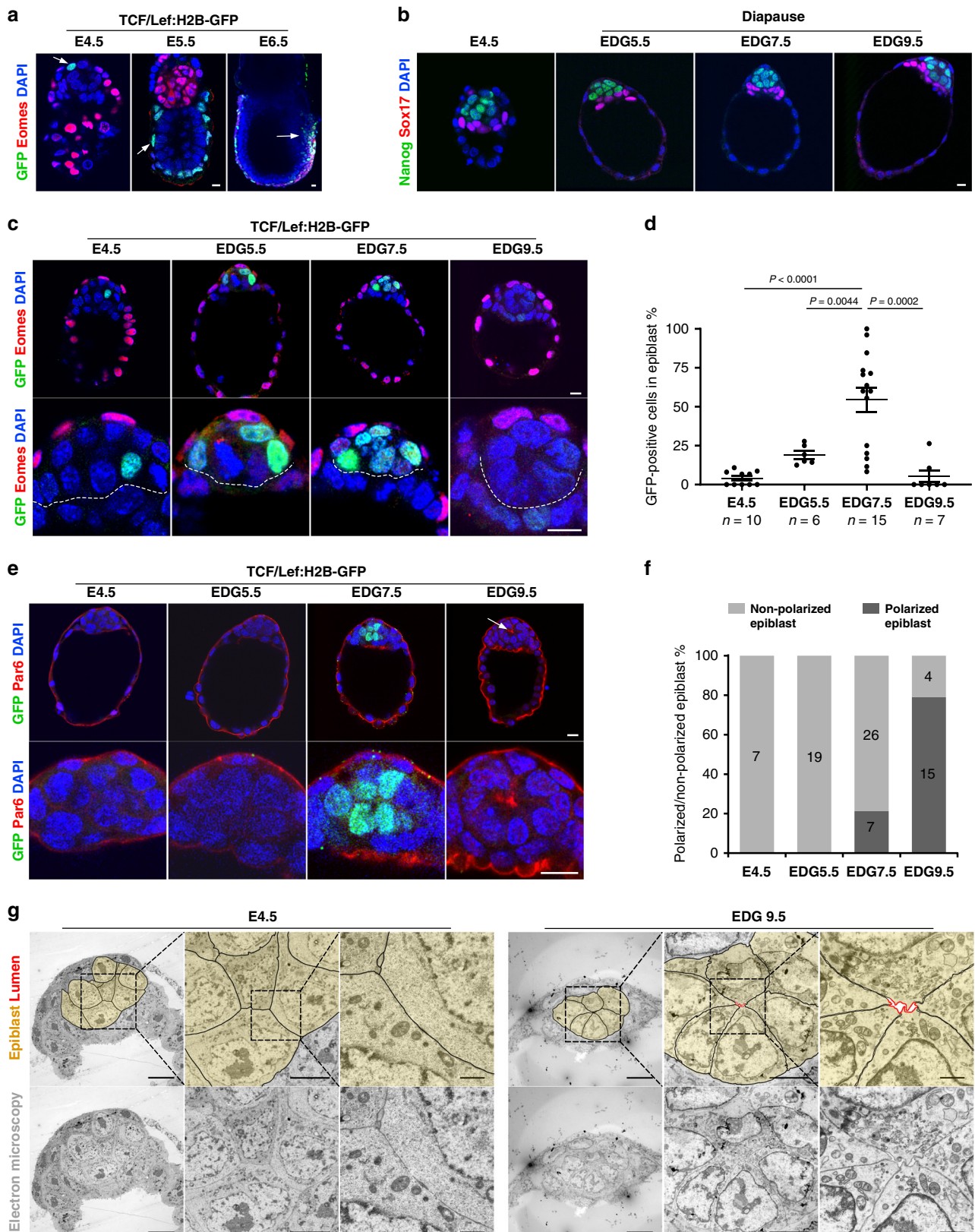

factors, such as Patj1 and Llgl2[30,39,40]. Spry2 belongs to a family of four mammalian Spry genes involved in the modulation of tyrosine receptor kinase signalling, such as Fgf, Egf and Vegf cascades[41]. Together with Spry4, Spry2 expression in the inner-cell mass of blastocyst-stage embryos was suggested to play a role in epiblast specification via opposing Fgf4-mediated signalling, which drives the primitive endoderm fate[42]. Thus, Spry2 may belong to a larger, yet unidentified network of effector proteins involved in both modulating cell fate specification and tissue morphology of the pluripotent lineage.

In vitro, the role of Wnt/β-catenin signalling in ES cells' self-renewal has been extensively studied. Inhibition of Gsk3 activity

**Fig. 5 Dynamics of Wnt signalling activity and epiblast self-organisation during diapause. a** TCF/Lef:H2B-GFP embryos isolated at E4.5, E5.5 and E6.5 and stained for GFP, Eomes and DAPI. Arrows indicate the location of the TCF/Lef:H2B-GFP-positive cells. **b** E4.5 blastocysts and EDG5.5, EDG7.5 and EDG9.5 diapause embryos stained for Nanog, Sox17 and DAPI. **c** TCF/Lef:H2B-GFP embryos isolated at E4.5, EDG5.5, EDG7.5 and EDG9.5 and stained for GFP, Eomes and DAPI. Dashed line indicates the border between the epiblast and the primitive endoderm. **d** Percentage of GFP-positive cells in the epiblast of E4.5, EDG5.5, EDG7.5 and EDG9.5 embryos. Data represent mean ± SEM, n = number of embryos, two-tailed unpaired Student's t test, the exact P value is noted in the figure. **e** TCF/Lef:H2B-GFP embryos isolated at E4.5, EDG5.5, EDG7.5 and EDG9.5 and stained for GFP, Par6 and DAPI. The apical domain in the epiblast of the EDG9.5 embryo is marked by an arrow. **f** Percentage of embryos with polarised and non-polarised epiblast at E4.5, EDG5.5, EDG7.5 and EDG9.5. The number of embryos is indicated on the graph. **g** Electron microscopy analysis of E4.5 and EDG9.5 blastocysts. The colour overlay (top panel) marks the epiblast (yellow) and the apical domain (red) that surrounds the central microlumen at EDG9.5. Scale bars **a–c**, **e**: 10 μm and **g** (from left to right): 10, 5 and 1 μm.

or genetic ablation of both Gsk3 α and β isoforms stabilises β-catenin, boosting the canonical Wnt signalling[12,43]. The subsequent interaction of β-catenin with the transcription repressor Tcf3 disrupts the Tcf3-mediated suppression of pluripotency factors, including Esrrb[11,24]. Accordingly, Tcf3-deficient ES cells show enhanced self-renewal, bypassing the requirement of CH stimulation[44,45]. In vivo, however, none of the previously reported Wnt signalling mutants exhibit defects in the pluripotent lineage before implantation[31]. β-catenin is required for initiation of gastrulation[32,33], whereas Tcf3-deficient embryos develop through gastrulation, forming expanded axial mesoderm structures[46]. However, normal embryonic development is a transient process, where the naive pluripotent cells only exist for a very short period of time before implantation. In contrast, in ES cells, the naive properties of the epiblast are captured and maintained over the long-term, potentially indefinitely.

Although in vivo the Wnt/β-catenin signalling is not essential for epiblast specification and development until the egg cylinder stage, we found that this pathway plays a role in the prolonged maintenance of the pre-implantation epiblast during diapause (Fig. 8g). We observed increasing Wnt reporter activity between EDG5.5 and EDG7.5 associated with non-polarised phenotype of the pluripotent lineage, similar to the effects of Wnt stimulation in 3D ES cell culture. By EDG9.5, the Wnt reporter activity decreased substantially and the epiblast cells reorganised into an epithelial rosette with central microlumen. Deletion of β-catenin resulted in a complete loss of pluripotent cells, whereas the extra-embryonic lineages were still maintained. The collapse of the blastocoel cavity in the mutant embryos during diapause could be the result of insufficient cell–cell adhesion, contributing to the severity of the phenotype. Accordingly, we observed a similar effect of decreased intracellular adhesion upon β-catenin depletion in 3D ES cell culture. In contrast, during transient development, the β-catenin null embryos do not exhibit an adhesion defect, as compensatory upregulation of plakoglobin (γ-catenin) expression takes over the role of β-catenin on the cell membrane[33]. This suggests that although both β-catenin signalling and adhesion functions are dispensable during normal embryogenesis, these functions are required in the blastocyst during diapause.

Another example of a pathway with an essential role in the maintenance of the pluripotent lineage during diapause is the Gp130 signalling. Although Gp130 function is critical for ES cells self-renewal in conventional Lif/serum culture conditions[47,48], gene ablation of Gp130 does not affect epiblast development in vivo. Gp130 knockout embryos die between E12 and E18 due to myocardial and neuronal defects[49,50]. However, this receptor is indispensable for the prolonged maintenance of the epiblast during diapause, as Gp130 deficiency results in loss of the pluripotent lineage in diapause embryos[51]. Thus, both Wnt and Gp130 signalling have cryptic functions in the epiblast that come into play only during embryonic diapause. Essentially, the responsiveness of ES cells to Wnt stimulation may have its physiological basis in diapause (Fig. 8g), as previously suggested for the Lif/Gp130 signalling[51].

In addition to the core pluripotency factors Oct4 and Sox2, ES cells express an "ancillary" network of transcription factors such as Esrrb, Nanog, Klf2, Klf4, Prdm14, Sall4, Tbx3, Tfcp2l1 and Nr0b1. These factors cross-regulate their expression via interconnected feedback loops and buffer the pluripotency circuit against cues that promote exiting the naive state[1,52]. However, in vivo, the genetic ablation of ancillary transcription factors has not been generally associated with defects in the naive epiblast. For instance, Nanog is required for primitive endoderm specification[53], whereas Esrrb knockout embryos exhibit placental abnormalities[34]. Moreover, the placental defects in Esrrb mutants were rescued via tetraploid complementation[34], indicating that Esrrb is not required for epiblast development during normal embryogenesis. Nevertheless, we found that during diapause, loss of Esrrb resulted in a marked decrease of epiblast cells. Similarly, other ancillary factors that are dispensable during transient development may play a role in the long-term maintenance of the pre-implantation epiblast during diapause. Thus, a number of such factors may function as a part of a fail-safe mechanism, sustaining the naive epiblast when embryonic development is put on hold.

We found that during diapause, the Wnt pathway is stimulated in an autocrine manner via embryo-produced Wnt ligands. In Wls-deficient blastocysts, where the endogenous Wnt ligand secretion is blocked, the TCF/Lef:H2B-GFP reporter remained inactive. At the same time, the Wls knockout embryos exhibited substantial loss of epiblast cells but preserved the blastocyst integrity, similar to the Esrrb mutants. Accordingly, embryos lacking Wls and Esrrb were less severely affected in comparison to embryos lacking β-catenin, where the blastocyst architecture was altered. Since the Wls knockout was global, not lineage-specific, this strategy cannot discriminate whether the source of Wnt ligands is the epiblast and/or the extra-embryonic tissues of the blastocyst. Previously reported transcriptional profiling of E4.5, diapause (EDG6.5) and E5.5 epiblasts shows upregulation of the canonical Wnt4 ligand during diapause[35]; although one cannot exclude the Wnt ligands being contributed by the trophectodermal and/or primitive endoderm, this suggests that Wnt pathway activation during diapause is an epiblast-intrinsic process.

Historically, the induction of diapause has been implicated as a tool that substantially increases the efficiency of ES cell derivation[9,10]. By combining diapause and Mek inhibitor treatment, ES cell derivation is enabled even from refractory strains such as CBA[54]. It is plausible that the activation of the Wnt pathway during diapause is a contributing factor, similar to the effects of Gsk3 inhibition in the context of the 2i medium, which enables the establishment of ES cell lines from any genetic background[12,55]. In addition, while our study was in revision, Neagu and colleagues reported that combined Wnt and Mek inhibition in the presence of Lif results in the stabilisation of E5.0

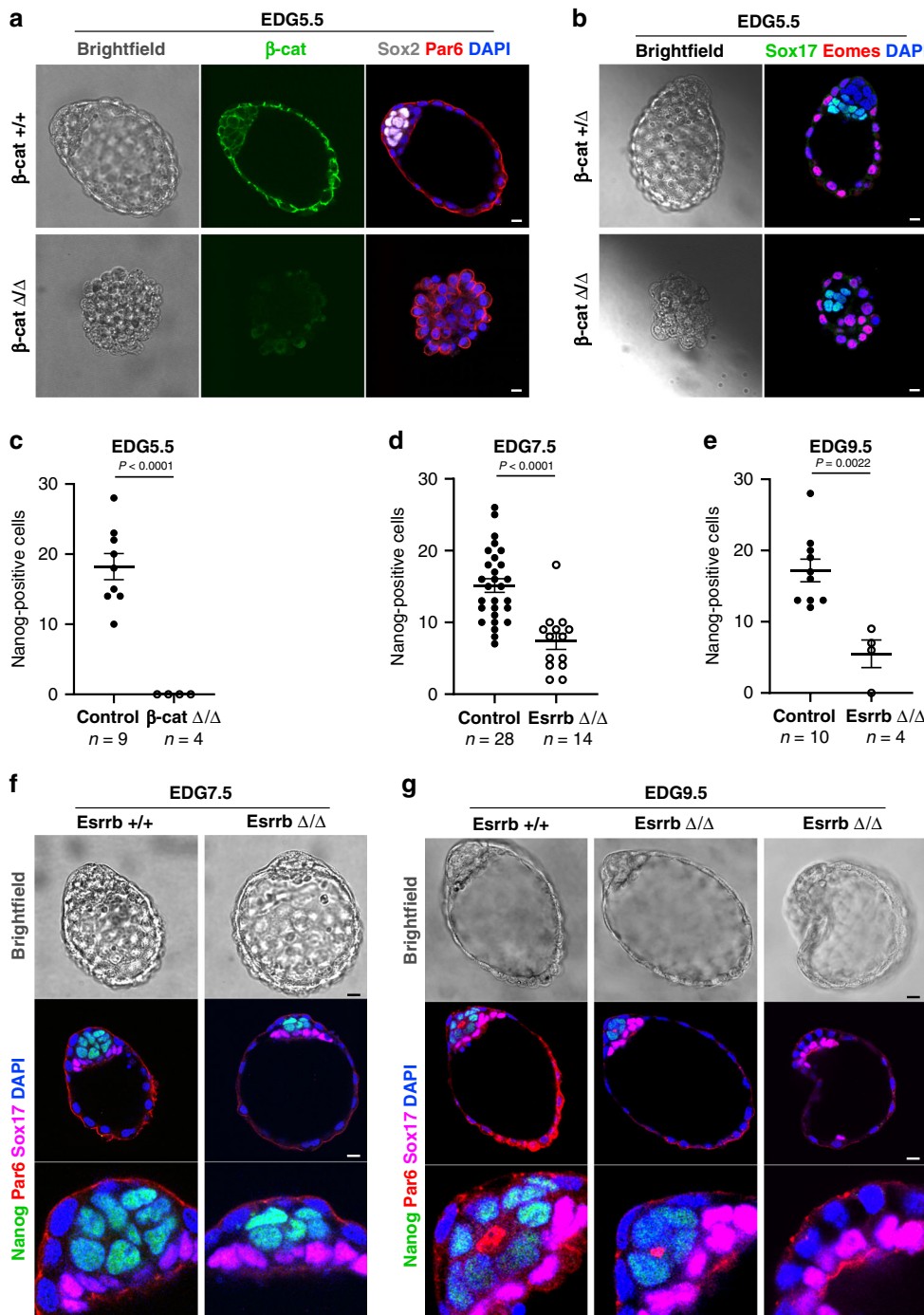

**Fig. 6 Loss-of-function analysis of β-catenin and Esrrb during diapause. a** β-catenin control (+/+) and knockout (Δ/Δ) embryos isolated at EDG5.5 and stained for β-catenin, Sox2, Par6 and DAPI. **b** β-catenin control (Δ/+) and knockout (Δ/Δ) embryos isolated at EDG5.5 and stained for Sox17, Eomes and DAPI. **c** The number of epiblast cells in β-catenin control (+/+ and +/Δ) and knockout (Δ/Δ) embryos at EDG5.5 based on Nanog expression. Data represent mean ± SEM, $n$ = number of embryos, two-tailed unpaired Student's $t$ test, the exact $P$ value is noted in the figure. **d** The number of epiblast cells in Esrrb control (+/+ and +/Δ) and knockout (Δ/Δ) embryos at EDG7.5 based on Nanog expression. Data represent mean ± SEM, $n$ = number of embryos, two-tailed unpaired Student's $t$ test, the exact $P$ value is noted in the figure. **e** The number of epiblast cells in Esrrb control (+/+ and +/Δ) and knockout embryos (Δ/Δ) at EDG9.5 based on Nanog expression. Data represent mean ± SEM, $n$ = number of embryos, two-tailed unpaired Student's $t$ test, the exact $P$ value is noted in the figure. **f** Esrrb control (+/+) and knockout (Δ/Δ) embryos isolated at EDG7.5 and stained for Nanog, Par6, Sox17 and DAPI. **g** Esrrb control (+/+) and knockout (Δ/Δ) embryos isolated at EDG9.5 and stained for Nanog, Par6, Sox17 and DAPI. Scale bars, 10 μm.

epiblast-like state in vitro (rosette pluripotency). These cells can revert to naïve pluripotency upon Wnt stimulation or progress into primed state upon Fgf/Erk activation, thus exhibiting naïve-prime intermediate be properties[56].

During diapause, the peak of Wnt stimulation at EDG7.5 was followed by a reduction of reporter activity by EDG9.5. What determines the dynamics of Wnt signalling in the epiblast? The autocrine Wnt stimulation in the first days of diapause could be

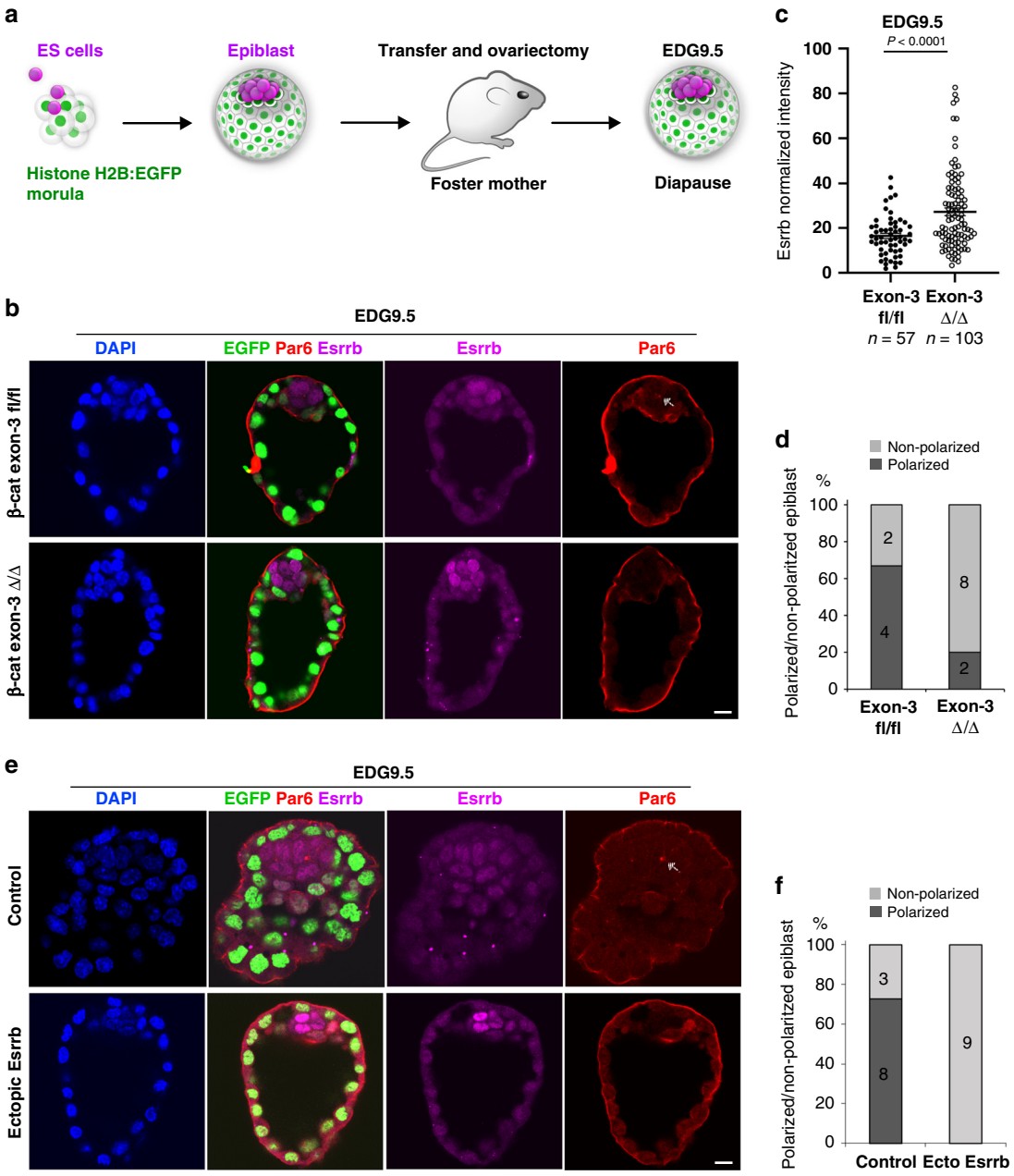

**Fig. 7 Wnt/β-catenin/Esrrb cascade controls the tissue-scale organisation of the epiblast during diapause. a** Schematic representation of the generation of chimeric blastocysts. **b** EDG9.5 chimeric embryos containing epiblast comprised of β-catenin exon-3 Δ/Δ or control β-catenin exon-3 fl/fl cells stained for Esrrb, EGFP, Par6 and DAPI. **c** Esrrb expression in β-catenin exon-3 Δ/Δ or control β-catenin exon-3 fl/fl epiblast cells at EDG9.5. Data represent mean ± SEM, $n$ = cells measured in ten β-catenin exon-3 Δ/Δ and six control β-catenin exon-3 fl/fl embryos, two-tailed unpaired Student's $t$ test, the exact $P$ value is noted in the figure. **d** Percentage of chimeric embryos with polarised and non-polarised β-catenin exon-3 Δ/Δ or control β-catenin exon-3 fl/fl epiblast. **e** EDG9.5 chimeric embryos containing epiblast comprised of wild-type (control) or Esrrb ectopically expressing ES cells stained for Esrrb, EGFP, Par6 and DAPI. **f** Percentage of chimeric embryos–control and ectopically expressing Esrrb, with polarised and non-polarised epiblast at EDG9.5. Scale bars, 10 μm.

an adaptive response promoting epiblast self-renewal. As the embryo gradually becomes biosynthetically dormant, the endogenous production of Wnt ligands may also get silenced, decreasing the Wnt pathway activity below the detection threshold of the reporter. Accordingly, the Wnt-repressive function that blocks epithelialization is discontinued after EDG7.5, allowing the reorganisation of the pluripotent cells into polarised rosettes with a central microlumen. Importantly, the role that this emerging tissue-scale architecture plays in the long-term maintenance of the epiblast is still an open question. In other model

systems, such as the Zebrafish lateral line neuromast, the luminal space in epithelial rosettes acts as a microenvironment that coordinates and enhances multicellular responses to signalling cues concentrated in the lumen[57]. Similarly, the rosette configuration of the epiblast may coordinate and reinforce intracellular communication to preserve the viability and developmental potential of the pluripotent lineage.

Altogether, our results show that while diapause embryos seemingly reside in a state of suspended animation, the diapause is not a "paused" state, but a dynamic process with underlying

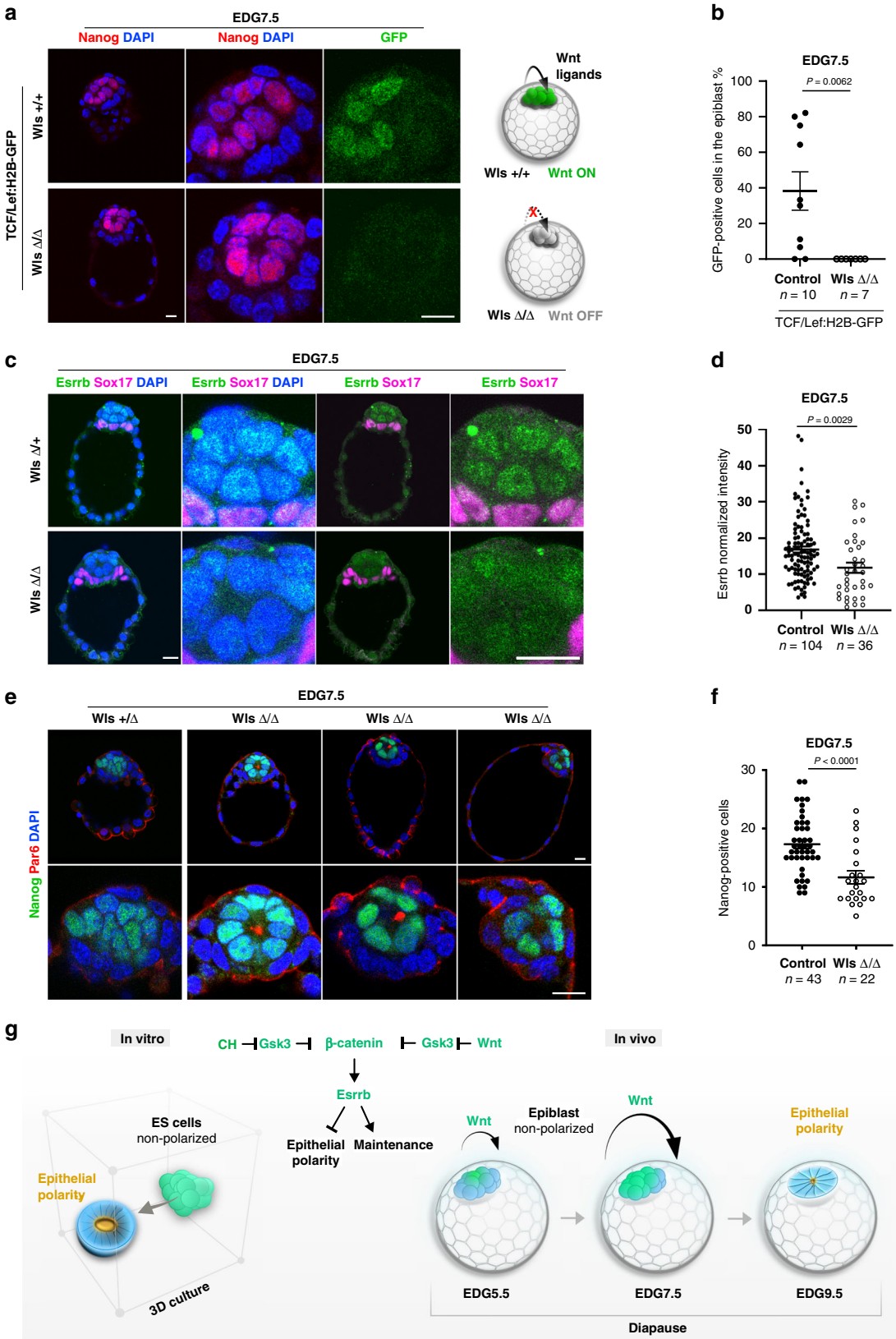

mechanisms that can appear redundant during transient embryonic development. Future research can reveal to what extent cell signalling and tissue organisation principles in the epiblast are shared in other mammalian species that utilise diapause as a reproductive strategy.

## Methods

**Cell lines**. Stable cell lines for constitutive transgene expression were generated using PiggyBac (PB) transposon system[58]. The transgenes were cloned into PB transposon vector (pIB10-PB-CAG-cHA-IRES-Puro) and co-transfected with transposase encoding plasmid (pIB1-PyCAG-PBase). After selection with puromycin (1 μg/ml), individual clones were picked and analysed by PCR, western blot

**Fig. 8 Wnt signalling is activated in an autocrine manner during diapause. a** Control TCF/Lef:H2B-GFP Wls +/+ and TCF/Lef:H2B-GFP Wls Δ/Δ embryos isolated at EDG7.5 and stained for GFP, Nanog and DAPI. Note the lack of GFP expression in the epiblast of Wls-deficient blastocysts. **b** Percentage of GFP-positive cells in epiblast in control TCF/Lef:H2B-GFP Wls (+/+ and +/Δ) and TCF/Lef:H2B-GFP Wls Δ/Δ embryos at EDG7.5. Data represent mean ± SEM, $n$ = number of embryos, two-tailed unpaired Student's $t$ test, ***$P$ < 0.001. **c** Wls control (+/Δ) and knockout (Δ/Δ) embryos stained for Esrrb, Par6 and DAPI. **d** Esrrb expression in the epiblast cells of Wls control (+/+ and +/Δ) and knockout (Δ/Δ) embryos at EDG7.5. Data represent mean ± SEM, $n$ = cells measured in eight Wls control and five knockout embryos, two-tailed unpaired Student's $t$ test, the exact $P$ value is noted in the figure. **e** Wls control (+/Δ) and knockout (Δ/Δ) embryos stained for Nanog, Par6 and DAPI. **f** The number of epiblast cells in Wls control (+/+ and +/Δ) and knockout (Δ/Δ) embryos at EDG7.5 based on Nanog expression. Data represent mean ± SEM, $n$ = number of embryos, two-tailed unpaired Student's $t$ test, the exact $P$ value is displayed in the figure. **g** Function of the Wnt/β-catenin/Esrrb cascade in vitro and in vivo. Scale bars, 10 μm.

and/or immunohistochemistry to identify transgene expression. The stable cell lines for inducible (Tet-on) transgene expression were also generated using the PB transposon system. The transgenes were cloned into pIB12-PB-hCMV1-cHA-IRES-Venus plasmid that contains tetracycline response elements and co-transfected with transposase (pIB1-PyCAG-PBase) and tetracycline-controlled transactivator (pIB11-PB-CAG-rtTAM2-IRES-Neo) plasmids, followed by selection with G418 (300 μg/ml). Tet-on transgene expression was stimulated using 1 μg/ml of Dox. Conditional ablation via Cre/loxP recombination was induced using 500 nM of 4OHT.

**Mice.** The mice used in this study were at the age of 6 weeks to 5 months. The animals were maintained under a 14-h light/10-h dark cycle with free access to food and water. Female mice were housed in groups of up to four per cage, and male stud mice were housed individually. Embryos for experiments were obtained from wild-type and transgenic strains from mattings using females with a natural ovulation cycle. Heterozygous mouse lines were generated from β-catenin fl/fl, Esrrb fl/fl and Wls fl/fl conditional knockout strains by crossings with Zp3-Cre mice. F1 generation of female mice with genotypes: β-catenin fl/+ Zp3-Cre, Esrrb fl/+ Zp3-Cre or Wls fl/+ Zp3-Cre were crossed with C57Bl/6 males to generate β-catenin +/−, Esrrb +/− or Wls +/− strains. Knockout embryos for experiments were derived from heterozygous intercrosses and compared to wild-type and heterozygous littermates. Animal experiments and husbandry were performed according to the German Animal Welfare guidelines and approved by the Landesamt für Natur, Umwelt und Verbraucherschutz Nordrhein-Westfalen (State Agency for Nature, Environment and Consumer Protection of North Rhine-Westphalia).

**Cell culture.** All ES cell lines were maintained in the presence of 2i/Lif (0.4 μM PD0325901; Cayman Chemical, 13034, 3 μM CHIR99021; Tocris, 4423 and 4 ng/ml Lif; prepared in house), on plastic dished coated with 0.2% gelatine and passaged at 1:6 ratio using 0.05% Trypsin-EDTA. The TCF/Lef:H2B-GFP ES cells were derived from TCF/Lef:H2B-GFP blastocyst. E3.5 embryos were plated into individual wells of a 96-well plate and cultured in DMEM medium supplemented with 15% FBS, 2 mM L-glutamine, 1 mM sodium pyruvate, 0.1 mM non-essential amino acids, 50 U/ml penicillin–streptomycin, 0.1 mM 2-mercaptoethnal, 0.4 μM PD0325901, 3 μM CHIR99021 and 4 ng/ml Lif. After 4 days of culture, the blastocyst outgrowths were dissociated using trypsin, transferred into fresh wells and further expanded once ES cell colonies emerged. TCF/Lef:H2B-GFP ES cells conversion to EpiLC was performed as follows. The TCF/Lef:H2B-GFP ES cells were grown on fibronectin-coated plates in N2B27 (N2; Thermo Fisher Scientific, 17502048, B27; Thermo Fisher Scientific, 12587001) medium supplemented with 2i/Lif for 2 passages. After that, the cells were plated on fresh fibronectin-coated plates and grown for 2 days in N2B27 medium supplemented with 1% KSR, 12 ng/ml Fgf2 and 20 ng/ml Activin A.

**3D cell culture.** The cells were dissociated using 0.05% trypsin-EDTA, pelleted by centrifugation (5 min/1000 rpm), then washed with PBS and repelleted. After removing any residual PBS, the cells were resuspended in Matrigel (Corning, 356231) (2000 cells per 1 μl of Matrigel), and 20 μl of the cell suspension was plated per single well of eight-well ibidi μ-plate. To solidify the hydrogel, the plate was placed for 2–3 min in a cell culture incubator at 37 °C. After that each well was filled with N2B27 medium supplemented with single or combination of inhibitors —0.4 μM PD0325901, 3 μM CHIR99021 or an equal volume of DMSO and ligands —12 ng/ml Fgf2, 20 ng/ml Activin A or 4 ng/ml Lif and cultured for 24 h, 36 h or 48 h at 37 °C and 5% $CO_2$ atmosphere in the air.

**Genomic DNA isolation and PCR genotyping.** Lysis buffer containing 50 mM KCl, 10 mM Tris-HCl pH 8.0, 2 mM $MgCl_2$, 0.45% NP-40, 0.45% Tween-20 and 0.4 mg/ml proteinase K was used to lyse tissues and extract DNA. The volume of the lysis buffer was adjusted based on the amount of material, 50 μl lysis buffer was used for earclips, and 10 μl lysis buffer was used for pre-implantation embryos. The tissue was lysed for at least 3 h at 55 °C, followed by 95 °C incubation for 15 min, and 1 μl of the solution was used for PCR genotyping with primers indicated in Supplementary Table 1.

**Western blot.** ES cells were lysed using buffer containing 10 mM Tris-HCl pH 7.6, 150 mM NaCl, 2 mM $MgCl_2$, 2 mM EDTA, 0.1% Triton-X-100, 10% glycerol and 1× protease inhibitors cocktail (cOmplete ULTRA). The lysate was kept on ice for 20 min and then sonicated for 5 min. Extracted proteins were loaded into PAA gel and transferred to the PVDF membrane followed by blocking in 5% dry milk in PBST for 30 min. The membrane was incubated with primary antibodies at 4 °C, overnight. On the next day, the membrane was washed with PBST and incubated with secondary antibodies conjugated to HRP for 2 h. The proteins were detected using ECL or ECL Plus and exposed to autoradiography films. Uncropped blots can be found in the source data file.

**Induction of diapause.** Diapause was induced via ovariectomy of female mice on the morning of day 3.5 post coitum. The pregnancy during diapause was maintained by a daily injection of 3 mg medroxyprogesterone 17-acetate (Depo Clinovir). Non-surgical diapause was induced via injection of 10 μg tamoxifen and 3 mg medroxyprogesterone 17-acetate at 1.5 and 2.5 d.p.c (days post coitum).

**Immunofluorescence.** Cells or embryos were fixed with 4% PFA for 20 min and washed twice using a washing solution of 1% FCS in PBS. The fixed samples were permeabilized with 0.3% Triton for 10 min and then washed twice. The primary antibody was applied in a blocking buffer of 2% FCS in PBS for 24 h at 4 °C. After three times of washing, samples were incubated in the secondary antibody solution for another 24 h at 4 °C, and the nuclei were counterstained with DAPI. The embryos were mounted on glass-bottom plates, in drops of 1% FCS/PBS under mineral oil, whereas ES cells were directly imaged in the ibidi μ-plates used for 3D culture. Primary antibodies and dilutions: mouse anti-Esrrb (R&D systems, PP-H6705-00, 1:300), mouse anti-Nanog (Cell signalling technology, 8822, 1:300), goat anti-Sox17 (R&D systems, AF1924, 1:300), mouse anti-Oct4 (Cell signalling technology, 83932, 1:300), rabbit anti-Pard6B1 (Santa Cruz Biotechnology, sc-67393, 1:300), mouse anti-Pard6B1 (Santa Cruz Biotechnology, sc-166405, 1:300), rabbit monoclonal anti-Sox2 (Cell signalling technology, 23064, 1:300), mouse anti-β-Catenin (BD Biosciences, 610154, 1:300), mouse anti-E-Cadherin (BD Biosciences, 610182, 1:300), goat anti-GFP (R&D systems, AF4240, 1:300), rat anti-Podocalyxin (R&D systems, MAB1556, 1:300), rabbit anti-Cleaved Caspase-3 (Cell Signalling Technology, 9664, 1:300), rabbit anti-Spry2 (Thermo, PA5-98172, 1:300), secondary antibodies and dilutions: Alexa 594 donkey anti-rabbit IgG (H+L) (1:200), Alexa 488 donkey anti-rabbit IgG (H+L) (1:200), Alexa 488 donkey anti-mouse IgG (H+L) (1:200), Alexa 647 donkey anti-rat IgG (H+L) (1:200), Alexa 647 donkey anti-mouse IgG (H+L) (1:200), Alexa 488 donkey anti-goat IgG (H+L) (1:200), Alexa 647 donkey anti-goat IgG (H+L) (1:200), Alexa 488 donkey anti-rabbit IgG (H+L) (1:200). F-actin was stained using Alexa Fluor® 647 Phalloidin. Images were acquired using Leica SP5 and Leica SP8 confocal microscopes and analysed by using Fiji software. Nuclear fluorescence intensity was normalised against the average background signal.

**Fluorescence-activated cell sorting (FACS).** Cells were dissociated using trypsin and transferred into PBS supplemented with 3% FCS. FACSAria IIu sorter (BD biosciences) was used for sorting and analysis. Single viable cells were first selected based on FSC and SSC gating, and then Venus-positive cells were collected. Flowjo software (TreeStar) was used for data analysis.

**Transmission electron microscopy.** Embryos were fixed in 2% PFA, 2% glutaraldehyde in 0.1 M cacodylate buffer, pH 7.4. For better handling, each sample was first embedded in 2% LMP-agarose and postfixed in 1% osmiumtetroxide, 1.5% potassiumferrocyanide in 0.1 M cacodylate buffer. Subsequently, the specimen was dehydrated stepwise in ethanol, including en bloc 0.5% uranyl acetate staining during 70% ethanol. The last dehydration step was performed in propylenoxide followed by epon embedding. Sectioning on an ultramicrotome (UC6, Leica) was done stepwise until the region of interest and 60-nm ultrathin sections were collected on formvar filmed one slot grid and counterstained with lead. The samples were imaged on a FEI-Tecnai 12-electron microscope at 80 kV (Thermofisher Scientific), and characteristic images were taken with a 2 K CCD-Veleta camera (EMSIS). For EDG9.5 embryos, a CLEM approach (correlative light electron microscopy) adapted from ref. [59] was applied to reach the area of interest (onset of

lumen formation). Mouse embryos were fixed in 4% PFA in 0.1 M PHEM buffer, pH 6.9 and stained overnight with phalloidin to visualise individual cells in the framework of lumen formation. The embryos were glued with agarose on a glass-bottom dish (ibidi) suitable for light microscopy and submerged with PBS. Subsequently, the coordinates/position in z of the lumen was determined using confocal imaging. After that, the embryos were fixed in 2% PFA, 2% glutaraldehyde in 0.1 M cacodylate buffer, pH 7.4 as described and processed further for epon embedding into a gelatine capsule. Target sectioning was performed until the calculated z position cross-sectioning the lumen initiation point.

**RNA sequencing and data analysis**. In all, 1 µg of the total RNA, with RIN (RNA integrity number) numbers above 7 (Agilent 2100 Bioanalyzer), were used. mRNA was enriched using NEBNext® Poly(A) mRNA Magnetic Isolation Module and cDNA library was prepared with NEBNext® Ultra™ II Directional RNA Library Prep Kit. Sequencing was performed on the NextSeq 500 system (75 cycles, high output, v2.5). The sequencing is performed in Core Facility Genomics of the Medical Faculty, University of Münster.

The RNA sequencing reads were aligned to the mouse genome mm10 with TopHat2 aligner (v2.1.1)[60] with default input parameters. The number of reads that were mapped to each Ensembl gene (GRCm38) was counted using the Python package HTSeq (v0.7.2)[60,61] with "htseq-count –stranded no". Principal component analysis and differential expression analysis was performed with raw counts using the R package DESeq2 (v1.18.1)[62]. Genes were considered as deregulated if |log2FC| > 1 and FDR < 0.01 using Benjamini–Hochberg multiple test adjustment[63]. Gene set enrichment analysis (GSEA) was performed by using GSEA software version 4.0.1[64] with default settings. Input files were adapted from normalised counts generated with VST function in DESeq2 package. In total, 1000 permutations were used to calculate P values based on the phenotype annotation.

**Quantitative PCR analysis**. The total RNA was extracted using the NucleoSpin RNA (MACHEREY-NAGEL) Mini kit for RNA purification. cDNA synthesis was performed using the M-MLV reverse transcriptase (MACHEREY-NAGEL). Transcript levels were detected using iTaq SYBR Green Supermix (Bio-Rad) with Quantstudio 3 (Applied Biosystems). Gene expression was normalised to the housekeeping genes Gapdh and calculated using the delta Ct algorithm. The primer sequences are listed in Supplementary Table S6.

**Analysis of publicly available data**. ChIP-seq data of Tcf3 and Esrrb were downloaded from GEO (GSE11724 and GSE11431, respectively). Input DNA or Mock IP samples from each study were used as spike controls. Sequencing reads are mapped to mm10 mouse genome reference with aligner STAR[65]. Peak calling was performed by MACS2[66] with setting "-g mm --nomodel --extsize 200 --SPMR -B -q 0.05". Normalised read coverage from MACS2 output was visualised with IGV[67].

**Statistics and reproducibility**. All the experiments presented in this manuscript are reproduced at least in three independent experiments. Images are shown as the representative of all independent experiments.

**Reporting summary**. Further information on research design is available in the Nature Research Reporting Summary linked to this article.

## Data availability
All RNA sequencing data are deposited in the NCBI GEO database under accession code: GSE141773. There is no restriction on data availability. Source data are provided with this paper.

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

## Acknowledgements
We thank Heike Brinkmann for the excellent technical assistance; Martin Stehling for the FACS analysis; Binyamin Düthorn, Niraimathi Govindasamy, Hatice Ozge Ozguldez, Anusha Sathyanarayanan, Fei Chen and Kirill Salewskij for helpful discussions; Hyun Woo Choi for providing E3 EpiSCs; Anika Witten for RNA-seq and Dr. Celeste Brennecka for proofreading the paper. This work was supported by the German Research Foundation (DFG) Emmy Noether grant (BE 5800/1-1) to I.B. and the Cells-In-Motion Cluster of Excellence (CiM) Pilot project grant (PP-2017-13) to R.F.

## Author contributions
I.B. and R.F. conceived the study, designed experiments and interpreted the results; R.F. performed most of the experiments. I.B. and R.F. wrote the paper; Y.S.K. generated and analysed ES cell lines; R.C. contributed to the analysis of diapause embryos; J.W. performed the bioinformatic analysis; K.M. and D.Z. performed the EM studies; K.A. and G.W. provided cell lines and mouse strains; S.G. generated and characterised cell lines; J.L. generated heatmaps of RNA-seq data; S.A.L. and H.R.S. provided support of key infrastructure, reagents and discussed the results.

## Funding

## Competing interests
The authors declare no competing interests.
