## [Peer Review File · Nature Communications]

Reviewers' Comments:

Reviewer #1:

Remarks to the Author:

Using a cell culture system described in their previous work, Bedzhov and collaborators report that canonical WNT signaling counteracts the establishment of apico-basal polarity in differentiating 3D aggregates, or "rosettes", derived from ESCs. The authors link the establishment of epithelial polarity to the transition from naïve to primed pluripotency, and propose *Esrrb* as the main target of WNT signaling in this context. The data presented indicates that forced expression of *Esrrb* blocks not only ESC differentiation, but also the establishment of cell polarity during rosette formation. The authors further identify in *Spry2* a downstream mediator of *Esrrb* activity, showing that its overexpression blocks the polarization of 3D aggregates. Assessing the developmental relevance of these observations, the authors confirm previous reports of low or absent canonical WNT signaling in the epiblast before implantation, but also uncover the activation of WNT pathway during diapause. They further show that the loss of β -catenin or autocrine WNT signals compromise the long-term maintenance of pluripotent cells in diapause embryos, a phenotype also observed after *Esrrb* inactivation. Finally, the data presented indicates that *Esrrb* null embryos show premature signs of epiblast polarization during diapause, coinciding with the loss of expression of naïve pluripotency markers.

This report significantly advances our understanding of the elusive function of WNT signaling and *Esrrb* during early murine development, highlighting a specific function during diapause. The study is overall well-constructed, presents clear data, and make use of elegant genetic approaches. Two main aspects remain to be improved: First, the experiments conducted during diapause fail to establish a clear causal connection between the activation of canonical WNT signaling, a sustained expression of *Esrrb*, and the inhibition of epiblast polarization (see 1/ below). Second, the transcriptional targets of WNT signaling during rosette formation, and the mechanistic mediators of *Esrrb* activity in opposing the establishment of cell polarity should be characterised more systematically (2/ below).

1/

The authors should present immunostainings indicating that *Esrrb* expression is diminished or ablated in β -catenin and *Wntless* deficient embryos during diapause. Similarly, an eventual loss of *Esrrb* should be linked to the altered expression of the proposed downstream effectors of this transcription factor in regulating polarization, at least by gene expression analysis. At the moment the data indicates that interfering with WNT signaling and *Esrrb* function compromises the maintenance of apolar and pluripotent epiblast cells during diapause, but fails to establish a connection between these observations.

2/

Fig 3D-F: It would be useful to provide a more detailed analysis of the gene expression changes observed after treating 3D cultures of ESCs with CH.

First, how many of the differentially expressed genes identified here were considered as possible targets of Tcf3? In Fig 3F the profiles of Tcf3 binding are shown in the regions immediately proximal to the gene bodies of a few selected targets. Yet, enhancers regulating expression of genes affected by CH are likely distributed over much greater distances. It would be important to present a more systematic analysis of the connection between the observed gene expression changes and Tcf3 binding. In this light it might also be useful to cross the data generated in this study with available gene expression profiles generated in *Tcf3*^{-/-} ESCs (Yi, ..., Merrill; Nature Cell Bio 2011).

Most importantly, the decision to focus the remaining analyses on few pluripotency transcription

factors, although justified by their role in ESCs, remains at the moment somehow arbitrary. The authors show that CH treatment does not inhibit the polarisation of 3D cultures from EpiLCs, which is used to support the conclusion that the activities inhibiting this process are specific to the naïve state. If this argument is to be made, the authors should discuss further the rationale of testing the ability of EpiSCs and EpiLCs to form polarized rosettes. While both naïve ESC and primed pluripotent cells are epithelial, the latter also show clear signs of polarity in culture. In this light, it would be important to determine whether the aggregates obtained soon after embedding primed pluripotent cells in matrigel are already polarized, as might be suggested by previous reports (Taniguchi,., Gumucio; Stem Cell Reports 2015). In this case, CH, rather than being unable to prevent polarization due to the lack of mediators expressed exclusively in the naïve state, would simply be operating on an already established polarized state. Also, while performing experiments involving CH treatment of EpiLCs or EpiSCs, the authors should consider that WNT activation, in contrast to what observed in naïve cells, can induce mesendodermal differentiation in primed pluripotent cells (Tsakiridis,., Wilson; Development, 2015 - Kemler,.,Solter; Development, 2004), a process which is linked to loss of epithelial character.

Overall, scrutinising more broadly the set of genes differentially expressed in response to CH might give additional insights. An attempt to a similar analysis is for instance presented in figure 4. Are any of these targets directly involved in the process of establishing cell polarity or cell adhesion? And if so, is the effect of CH treatment, at least in part, specific to the context in which the analysis was performed? In this sense, it would be helpful to understand how the effects of CH treatment in the 3D culture system used by the authors relate to the effects observed in conventional culture conditions, for which published datasets are available (Dunn,., Martello; EMBO J., 2018).

Figure 4: As in figure 3, the link between gene expression changes and Esrrb binding should be explored more systematically. Similarly, the selection of the candidates to be characterized in functional experiments should be justified more thoroughly. Intersecting the gene expression changes observed after CH treatment in WT ESC and after Esrrb knockout (datasets used in figure 3 and 4, as done for few genes in Fig S3B and 4G) represents a possible initial approach.

Minor remarks:

1/

Figure S2D-E: Technical comment. Please provide gene expression data to match the analysis of protein levels, and present data for Klf2, Tfcp2l1 and Dax1 overexpression. In addition, it would be important to monitor the expression levels of each factor along the 2 days of 3D culture, as the output of the transgenes might be affected by differentiation. At the moment it is difficult to exclude that the lack of phenotype observed after overexpression of factors other than Esrrb is second to low/variable levels of overexpression.

Figure 4J: As above, the levels of expression of each factor should be reported before and throughout 3D culture.

2/

Line 238: The authors might consider mentioning that LIF has been proposed to act in parallel to Esrrb in maintaining naïve pluripotency (Martello,., Smith; Cell Stem Cell 2012). Most importantly, the notion that the loss of LIF signaling, while not displaying overt pre-implantation phenotypes, compromises the maintenance of pluripotent cells in the epiblast during diapause should be later discussed in relation to the results presented in the manuscript (Nichols,., Smith; Development 2001).

3/

In parallel to showing that naive pluripotency transcription factors repress polarization, the authors could consider discussing the possibility that the maintenance of an apolar character might be reciprocally promoting permanence in the naïve state. For instance, inhibition or loss of PKCi has been shown to impair the differentiation of ESCs (Mha,..., Mariani; Stem Cell Reports 2015, Dutta,..., Paul; Stem Cells 2011), and promote the acquisition of traits peculiar to naïve pluripotency in primed hESCs (Takashima,..., Smith; Cell 2014).

4/

Fig 4B suggests that CH treatment might be exacerbating cell death in *Esrrb* knockout ESCs. Are these images representative of all experiments performed?

5/

Line 258: ESCs cultured in 2i/LIF can self-renew in the absence of *Esrrb* (Martello,..., Smith; Cell Stem Cell 2012). Is the reduced viability reported by the authors specifically observed during differentiation in 3D cultures, or also in conventional 2i/LIF cultures?

6/

The authors could consider discussing the potential role of *Esrrb* in regulating rosette formation and cavitation in the extraembryonic ectoderm (Christodoulou,..., Zernicka-Goetz; Nature Cell Biology 2018), where *Esrrb* remains expressed after implantation.

Reviewer #2:

Remarks to the Author:

The authors in this study investigated the signaling pathway to sustain the naïve pluripotency of epiblast cells. They show that active Wnt/ β -catenin signaling is critical to suppress the differentiation/epithelialization of embryonic stem cells (ES) in a 3D in vitro culture model. *Esrrb*, a downstream target of β -catenin, mediates the suppression of epithelialization as long as ES cells are in naïve pluripotent status. Although the in vivo roles of β -catenin and *Esrrb* during epiblast epithelialization is limited in normal pregnancy, they found that Wnt/ β -catenin signaling is active in epiblast of dormant embryos during diapause. Using Wntless null embryos, which are deficient in secretion of ligands, the authors suggest that Wnt ligands were produced by diapausing embryos but not the uterus.

The mechanism of epiblast differentiation and loss of naïve pluripotency is not clearly understood. The investigators used genetic ES cell models and 3D in vitro culture system to mimic epiblast epithelialization. The identification of Wnt/ β -catenin signaling in suppressing epiblast differentiation is important. The manuscript is well organized and written.

Major concern is about the in vivo role of the Wnt/ β -catenin signaling in epiblast epithelization. The data suggest that the Wnt/ β -catenin signaling has a limited role in epiblast epithelization during normal pregnancy. Although Wnt/ β -catenin signaling is active in epiblast during diapause, the status of epiblast cells in diapause is different from that in the in vitro cultures and normal pregnancy. After the epithelization of epiblast cells in dormant embryos, the epiblast cells still maintain naïve pluripotency (Nanog positive, Figure 5b). However, in culture, ES cells lose Nanog while undergoing epithelization (Figure 1d). These results raise the question whether epiblast naïve pluripotency and its epithelization are mutually exclusive. Does the Wnt/ β -catenin signaling regulated cell polarity determine cell pluripotency in diapause embryos? Given the length of diapause which could be months in specific species, it may be questioned whether the embryo quality affects epithelization in the epiblast.

In line 63, the authors stated that "In mice, by embryonic day four and a half (E4.5), the

specification of these three lineages is completed, establishing a mature blastocyst. After the E4.5 embryo hatches from its glycoprotein envelope (Zona pellucida), it comes into direct contact with the mother and initiates implantation into the uterine wall." The pregnancy timing described in the paragraph is not correct. Blastocysts hatch on E3.5, and implant in the midnight of day 4 of pregnancy. Day 1 is considered finding a vaginal plug in the morning. The reference cited for this time-line should be replaced by more appropriate citation.

The authors stated that "a temporary spike in the levels of ovarian estrogen at day 4 of pregnancy leads to the removal of this protective layer, establishing a limited receptive phase". Embryos downregulate maternal MUC1 expression. However, preimplantation estrogen is not an absolute requirement for zona dissolution. They suggest Esrrb is downstream of β -catenin, but did not provide direct evidence. In figure 3h, an experiment of overexpression of Esrrb in β -catenin deleted ES could prove Esrrb is downstream of β -catenin.

In figure S2f, how could the authors differentiate trophoblast cells and epiblast cells?

In figure 4g and 4j, given Ntn1 and Spry2 decrease in Esrrb null, the deletion of Spry2 in ES cells could show more phenotype? Figure 4i showed some effect of Spry2 overexpression on cell polarization, suggesting that Spry2 may play a role during the process. Whether Esrrb controls the epithelial program via Spry2 is still an open question.

Reference 33 used embryos from EDG6.5 but not 7.5.

In figure 5e, the quality of the dormant embryo on EDG5.5 is not of high caliber.

The model of Wntless is not a clean model, since the Wntless null embryos are not able to release any ligands, not limited to Wnt ligands. The interaction between embryos and uteri is mutual. The model could stop the secretion from embryos to uteri, and indirectly interferes the maternal secretion.

Since diapause occurs within the uterus, not in in vitro settings, epiblast epithelialization must involve factors from the uterus and/or withdrawal of some signals emanating from the uterus. An in vitro model to induce blastocyst diapause could answer many of the concerns.

Reviewer #1 (Remarks to the Author):

Using a cell culture system described in their previous work, Bedzhov and collaborators report that canonical WNT signaling counteracts the establishment of apico-basal polarity in differentiating 3D aggregates, or “rosettes”, derived from ESCs. The authors link the establishment of epithelial polarity to the transition from naïve to primed pluripotency, and propose *Esrrb* as the main target of WNT signaling in this context. The data presented indicates that forced expression of *Esrrb* blocks not only ESC differentiation, but also the establishment of cell polarity during rosette formation. The authors further identify in *Spry2* a downstream mediator of *Esrrb* activity, showing that its overexpression blocks the polarization of 3D aggregates. Assessing the developmental relevance of these observations, the authors confirm previous reports of low or absent canonical WNT signaling in the epiblast before implantation, but also uncover the activation of WNT pathway during diapause. They further show that the loss of β -catenin or autocrine WNT signals compromise the long-term maintenance of pluripotent cells in diapause embryos, a phenotype also observed after *Esrrb* inactivation. Finally, the data presented indicates that *Esrrb* null embryos show premature signs of epiblast polarization during diapause, coinciding with the loss of expression of naïve pluripotency markers.

This report significantly advances our understanding of the elusive function of WNT signaling and *Esrrb* during early murine development, highlighting a specific function during diapause. The study is overall well-constructed, presents clear data, and make use of elegant genetic approaches. Two main aspects remain to be improved: First, the experiments conducted during diapause fail to establish a clear causal connection between the activation of canonical WNT signaling, a sustained expression of *Esrrb*, and the inhibition of epiblast polarization (see 1/ below). Second, the transcriptional targets of WNT signaling during rosette formation, and the mechanistic mediators of *Esrrb* activity in opposing the establishment of cell polarity should be characterised more systematically (2/ below).

We thank the reviewer for his/her positive comments. We are pleased that the reviewer considers that our work significantly advances the understanding of the Wnt signaling and *Esrrb* during early embryogenesis and we are grateful for his/her constructive suggestions that helped to improve the manuscript. We have addressed all his/her comments in full below.

1/

The authors should present immunostainings indicating that *Esrrb* expression is diminished or ablated in β -catenin and *Wntless* deficient embryos during diapause. Similarly, an eventual loss of *Esrrb* should be linked to the altered expression of the proposed downstream effectors of this transcription factor in regulating polarization, at least by gene expression analysis. At the moment the data indicates that interfering with WNT signaling and *Esrrb* function compromises the maintenance of apolar and pluripotent epiblast cells during diapause, but fails to establish a connection between these observations.

We agree with Reviewer 1 comments and following his/her request, we conducted series of experiments to examine and establish stronger connection between Wnt and *Esrrb* functions in the epithelialization and maintenance of the epiblast during embryonic diapause.

As a starting point, we analyzed *Esrrb* expression dynamics and we found that *Esrrb* expression decreases at EDG9.5 in comparison to EDG7.5 (new Figures S7c and S7d), which correlates with the dynamics of the Wnt signaling activity during diapause (high at EDG7.5 and low EDG9.5, Figures 5c and 5d). Next, we designed several experiments to examine the causal link between Wnt signaling, *Esrrb* expression and epithelial polarity.

First, we asked whether active Wnt signalling can sustain high *Esrrb* expression levels in the epiblast during diapause. We found that in vitro (in 3D ES cell culture) stabilizing β -catenin following

exon-3 deletion resulted in sustained *Esrrb* expression, in comparison to control exon-3 floxed ES cells (new Figure S3d). Next, we implement this system to maintain Wnt activity in context of the diapause embryo. Briefly, we aggregated b-cat exon-3 deleted or control exon-3 floxed ES cells with host morulae expressing Histone-H2B:EGFP reporter allele. The resulting chimeric blastocysts comprised of wild-type (EGFP+) extraembryonic tissues and EGFP- epiblast comprised of the donor cells. These embryos were transferred into foster mother, where diapause was induced via ovariectomy and subsequently recovered at EDG9.5 (new Figures 7a and 7b). We found that embryos where b-catenin was stabilized displayed higher levels of *Esrrb* expression and low rate of epiblast polarization (new Figures 7b – 7d), compared to control embryos. This indicates that active canonical Wnt/b-cat signaling maintains *Esrrb* expression during diapause and counters epithelialization of the pluripotent lineage.

Second, we asked whether sustaining the expression of *Esrrb* alone (without experimentally modulating Wnt activity) is sufficient to block the establishment of epithelial polarity at EDG9.5. Using the same strategy, we aggregated ES cells that constitutively express *Esrrb* transgene or control wild-type ES cells with host morulae (EGFP+) to generate chimeric blastocysts. The embryos were transferred into foster mothers, where diapause was induced via ovariectomy, and subsequently recovered at EDG9.5. Control embryos exhibited epiblast polarization, whereas the pluripotent lineage of embryos ectopically expressing *Esrrb* remained non-polarized (new Figures 7e and 7f). We also attempted to examine the expression of *Srpy2*, however the antibody sensitivity was below the threshold of the endogenous protein and the signal was only detectable upon *Srpy2* overexpression in 3D culture (new Figure S4f).

Finally, we examined *Esrrb* expression in the epiblast upon loss of Wnt signalling activity. Since, b-catenin knock out embryos exhibit a complete loss of the pluripotent lineage and therefore loss of epiblast cells required for the readout, we used Wls mutants to analyze the effects on *Esrrb* expression. We found that at the peak of Wnt activity (EDG7.5), *Esrrb* was clearly detectable in the epiblast of control (wild-type and Wls +/-del) embryos, whereas *Esrrb* expression was diminished in Wls knockout embryos (new Figures 8c and 8d). This is in accord with the epithelialization of the pluripotent lineage and decrease in the number of epiblast cells that we observed in Wls deficient blastocyst at EDG7.5 (Figure 8e). Taken together, this new experimental data, together with the original results of the analysis in Wls, b-cat and *Esrrb* knock out embryos and the in-depth analysis in 3D ES cell culture, provide strong evidence that Wnt/b-cat/*Esrrb* signaling controls the tissue-scale reorganization and maintenance of the epiblast during diapause. To further emphasize on these findings, we updated the title of the manuscript, as follows “Wnt/Beta-catenin/*Esrrb* signalling controls the tissue-scale reorganization and maintenance of the pluripotent lineage during embryonic diapause”.

2/

Fig 3D-F: It would be useful to provide a more detailed analysis of the gene expression changes observed after treating 3D cultures of ESCs with CH.

First, how many of the differentially expressed genes identified here were considered as possible targets of Tcf3? In Fig 3F the profiles of Tcf3 binding are shown in the regions immediately proximal to the gene bodies of a few selected targets. Yet, enhancers regulating expression of genes affected by CH are likely distributed over much greater distances. It would be important to present a more

systematic analysis of the connection between the observed gene expression changes and Tcf3 binding. In this light it might also be useful to cross the data generated in this study with available gene expression profiles generated in Tcf3^{-/-} ESCs (Yi, ..., Merrill; Nature Cell Bio 2011).

To identify Wnt target genes that suppress the establishment of epithelial polarity, we compared the transcriptomes of CH- versus DMSO-treated ES cells grown in 3D culture conditions for 48 h (Figures 3d, 3e and S2c; Table S1). Using available Tcf3 ChIP-seq data (Marson, A. et al. Cell 2008) and Tcf3 knock out RNA-seq data (Yi, F. et al. Nat Cell Biol 2011), we considered only Tcf3-bound genes, which expression was upregulated upon CH-treatment and Tcf3 depletion, as potential candidates. We found 52 genes that met these criteria (new Figure S2e, new Table S3 and updated main text), but examining all these genes individually is a brute force approach that is not practically feasible. Importantly, our analysis showed that active Wnt signalling counters epithelialization only in ES cells, but not in primed cells (EpiLC and EPI-SC), suggesting that the downstream factors mediating this effect are part of the naïve pluripotency network. Therefore, we considered only naïve pluripotency factors. We focused on the Nr0b1, Nanog, Tfc2l1, Klf2 and Esrrb as these factors were previously established as the key Wnt associated genes involved in naïve pluripotency (Martello, ..., Smith; Cell Stem Cell 2012). We identified Esrrb as an essential regulator of polarity and continued with the further in-depth analysis in vitro and in vivo.

Most importantly, the decision to focus the remaining analyses on few pluripotency transcription factors, although justified by their role in ESCs, remains at the moment somehow arbitrary. The authors show that CH treatment does not inhibit the polarisation of 3D cultures from EpiLCs, which is used to support the conclusion that the activities inhibiting this process are specific to the naïve state. If this argument is to be made, the authors should discuss further the rationale of testing the ability of EpiSCs and EpiLCs to form polarized rosettes. While both naïve ESC and primed pluripotent cells are epithelial, the latter also show clear signs of polarity in culture. In this light, it would be important to determine whether the aggregates obtained soon after embedding primed pluripotent cells in matrigel are already polarized, as might be suggested by previous reports (Taniguchi, ..., Gumucio; Stem Cell Reports 2015). In this case, CH, rather than being unable to prevent polarization due to the lack of mediators expressed exclusively in the naïve state, would simply be operating on an already established polarized state. Also, while performing experiments involving CH treatment of EpiLCs or EpiSCs, the authors should consider that WNT activation, in contrast to what observed in naïve cells, can induce mesendodermal differentiation in primed pluripotent cells (Tsakiridis, ..., Wilson; Development, 2015 - Kemler, ..., Solter; Development, 2004), a process which is linked to loss of epithelial character.

We share the same opinion as Reviewer 1 that EpiLC aggregates are already polarized. This was stated later in the original text *"we converted EpiLC back to ES cells and asked whether Esrrb can also inhibit established polarity during the reprogramming to naïve pluripotency."*

In addition, following the request of Reviewer 1, we cultured EpiLC in matrigel, alongside ES cells, and examined the establishment of epithelial polarity at 12h, 24h and 48h of 3D culture in the presence of DMSO or CH. First, we confirmed that EpiLC are already polarized (new Figure S1d). Second, we analyzed the effects of CH treatment using mesendodermal and pluripotency markers and found, as expected, that Wnt activation induced mesendodermal differentiation (new Figure S1e). As this is an early differentiation response, we did not observe loss of epithelial phenotype and induction of EMT (epithelial to mesenchymal transition), within the 48h time-frame of treatment. We also updated the main text as follows *"The Wnt reporter was also activated in TCF/Lef:H2B-GFP EpiLC, but, in*

contrast to ES cells, the EpiLC remained polarized in the presence of the Gsk3 inhibitor". We added that "Further analysis at 12h, 24h and 48h of 3D culture showed that EpiLC are epithelial cells and CH treatment in this context induces the expression of early mesendodermal markers, in agreement with the previously reported Wnt function in cell differentiation ²¹."

It is worth mentioning that loss of polarity was observed upon reprogramming of EpiLC to ES cells via ectopic expression of *Esrrb* in the presence of *Lif* (Figures 3j-3l). This reprogramming experiment indicated that *"reinstating the naïve state is a prerequisite for dismantling already established epithelial polarity in more developmentally advanced pluripotent cells."*

Overall, scrutinising more broadly the set of genes differentially expressed in response to CH might give additional insights. An attempt to a similar analysis is for instance presented in figure 4. Are any of these targets directly involved in the process of establishing cell polarity or cell adhesion? And if so, is the effect of CH treatment, at least in part, specific to the context in which the analysis was performed? In this sense, it would be helpful to understand how the effects of CH treatment in the 3D culture system used by the authors relate to the effects observed in conventional culture conditions, for which published datasets are available (Dunn, ..., Martello; EMBO J., 2018).

Looking more broadly on the gene expression changes we performed GSEA (gene set enrichment analysis) of CH and DMSO treated samples. Consistent with the establishment of epithelial polarity in the absence of Wnt activation, gene set enrichment analysis showed an increment of focal adhesion, adherens and tight junction expression in DMSO treated cells (new Figure S2d). In general, focal, adherens and tight junction adhesive factors serve as structural and mechanosensitive components in epithelial tissues. The activation of these genes is in accord with the de novo establishment of epithelial polarity in ES cells, as the cells exit naïve pluripotency. Keeping this in mind, we aimed to find key players (such as *Esrrb*) that interconnect pluripotency transitions and epithelialization. Importantly, the 3D culture approach is closest system currently available that mimics epiblast morphogenesis in vivo. In our own experience, the process of epithelialization in conventional (2D) settings is sporadic. This is most likely because the plastic surface of the culture plates, does not provide sufficient polarization cues, compared to the 3D extracellular matrix. In addition, plastic (including coated plastic) is a very stiff substrate that often results in cells spreading instead of forming regular epithelium. Moreover, the 2D culture does not provide the proper conditions for the establishment of lumen that can potentially further enforce the epithelial phenotype, as shown in other systems (Durdu ..., Gilmour, Nature 2014).

Figure 4: As in figure 3, the link between gene expression changes and *Esrrb* binding should be explored more systematically. Similarly, the selection of the candidates to be characterized in functional experiments should be justified more thoroughly. Intersecting the gene expression changes observed after CH treatment in WT ESC and after *Esrrb* knockout (datasets used in figure 3 and 4, as done for few genes in Fig S3B and 4G) represents a possible initial approach.

In order to identify the downstream effectors of *Esrrb* that control the establishment of apical-basal polarity in a Wnt dependent manner, we examined the genes differentially expressed upon *Esrrb* depletion and intersect this list with genes which modulate their expression in CH vs DMSO treated samples. Using available *Esrrb* ChIP-seq data (Chen ... Ng, Cell 2008), we considered only *Esrrb* bound

genes. We end up with a total number of 442 genes that met these criteria (new Figure S4c, new Table S5). We focused on candidates with a previously assigned regulative or structural role in epithelial tissues (as examining the full list is not practically feasible). After a careful (and laborious) literature mining, we shortlisted *Arl4c*, *Krt18*, *Ntn1* and *Spry2*. For each of these genes we provided a description to justify the subsequent functional experiments. “*Arl4c (ADP-ribosylation factor-like 4c) is a small GTPase that acts as a molecular switch promoting the activation of Cdc42, which is a central factor orchestrating apical-basal polarity in epithelial cells* ²⁷. *Krt18 (Keratin 18) forms intermediate filaments that are essential cytoskeletal components of polarized epithelia* ²⁸. *Ntn1 (Netrin-1) controls Par complex localization during axon guidance* ²⁹, and *Spry2 (Sprouty2) is a modulator of tyrosine receptor kinase signalling that was previously shown to repress the polarized epithelial phenotype of cancer cells* ³⁰.”

Minor remarks:

1/

Figure S2D-E: Technical comment. Please provide gene expression data to match the analysis of protein levels, and present data for *Klf2*, *Tfcp2l1* and *Dax1* overexpression. In addition, it would be important to monitor the expression levels of each factor along the 2 days of 3D culture, as the output of the transgenes might be affected by differentiation. At the moment it is difficult to exclude that the lack of phenotype observed after overexpression of factors other than *Esrrb* is second to low/variable levels of overexpression.

We provide qPCR analysis of *Tfcp2l1*, *Klf2*, *Nanog*, *Nr0b1* and *Esrrb* expression compared to control wild-type ES cells in conventional 2i/Lif culture conditions (0h) and at 48h of 3D culture in CH or DMSO supplemented medium (new Figure S3c), which shows proper ectopic expression of all factors, including *Esrrb*.

Figure 4J: As above, the levels of expression of each factor should be reported before and throughout 3D culture.

Similarly, we provide qPCR analysis of *Arl4c*, *Krt18*, *Ntn1* and *Spry2* expression compared to control wild-type ES cells in conventional 2i/Lif culture conditions (0h) and at 48h of 3D culture in CH or DMSO supplemented medium (new Figure S4e), which shows proper ectopic expression of all factors. In addition, we validated *Spry2* expression on protein level (new Figure S4f).

2/

Line 238: The authors might consider mentioning that LIF has been proposed to act in parallel to *Esrrb* in maintaining naïve pluripotency (Martello,..., Smith; Cell Stem Cell 2012). Most importantly, the notion that the loss of LIF signaling, while not displaying overt pre-implantation phenotypes, compromises the maintenance of pluripotent cells in the epiblast during diapause should be later discussed in relation to the results presented in the manuscript (Nichols,..., Smith; Development 2001).

We are grateful to Reviewer 1 for his/her suggestions. Matrello et al indeed suggested that LIF can act in parallel to *Esrrb*. This experiment was performed using (short term) siRNA mediated knockdown of *Esrrb*. In our experience *Esrrb* genetic ablation in ES cells is detrimental even in the presence of LIF. Three of the co-author in our manuscript (Kenjiro Adachi, Guangming Wu and Hans R.

Schöler) previously published a study (Adachi ... Schöler, Cell Stem Cell, 2018) reporting that Esrrb is required for ES cells maintenance in both Lif or ground state (2i/Lif) culture conditions.

[Redacted]

(Figure 1A and 1B from Adachi ... Schöler, Cell Stem Cell, 2018)

The second point that the Reviewer 1 raised, about Lif/Gp130 signaling during diapause is very important and following Reviewer 1 recommendation we included the following paragraph in the Discussion.

“Another example of a pathway with essential role in the maintenance of the pluripotent lineage during diapause is the Gp130 signalling. Although Gp130 function is critical for ES cells self-renewal in conventional Lif/serum culture conditions ^{47,48}, gene ablation of Gp130 does not affect epiblast development in vivo. Gp130 knockout embryos die between E12-E18 due to myocardial and neuronal defects ^{49,50}. However, this receptor is indispensable for the prolonged maintenance of the epiblast during diapause, as Gp130 deficiency results in loss of the pluripotent lineage in diapause embryos ⁵¹. Thus, both Wnt and Gp130 signalling have cryptic functions in the epiblast that come into play only during embryonic diapause. Essentially, the responsiveness of ES cells to Wnt stimulation may have its physiological basis in diapause (Figure 8g), as it was previously suggested for the Lif/Gp130 signalling ⁵¹.”

3/

In parallel to showing that naive pluripotency transcription factors repress polarization, the authors could consider discussing the possibility that the maintenance of an apolar character might be reciprocally promoting permanence in the naïve state. For instance, inhibition or loss of PKCi has been shown to impair the differentiation of ESCs (Mha,..., Mariani;Stem Cell Reports 2015, Dutta,..., Paul; Stem Cells 2011), and promote the acquisition of traits peculiar to naïve pluripotency in primed hESCs (Takashima,..., Smith; Cell 2014).

We appreciate Reviewer 1 suggestion and we agree that there is a good amount of evidence indicating that atypical PKC play a role in human and mouse ES cells pluripotency. What exact role mechanistically PKC play and how PKC inhibitor lead to the described effects is still elusive. Mha,..., Mariani;Stem Cell Reports 2015 suggest that Notch signalling is involved and Takashima,..., Smith; Cell 2014 hypothesize that epithelial polarity is suppressed, but also state that “The mechanism downstream of aPKC inhibition remains to be elucidated”. Unfortunately, none of these studies directly examines epithelial polarity and therefore we feel that more work has to be done in this aspect, before we can be confident that loss of polarity can sustain naïve characteristics.

4/

Fig 4B suggests that CH treatment might be exacerbating cell death in Esrrb knockout ESCs. Are these images representative of all experiments performed?

Yes, Esrrb deletion is detrimental for cells in both DMSO and CH supplemented medium, with stronger disadvantage (increased apoptosis) for the cells when cultured with CH. This effect is clearly visible during the second day of culture, as indicated by Figure 4B.

5/

Line 258: ESCs cultured in 2i/LIF can self-renew in the absence of Esrrb (Martello,..., Smith; Cell Stem Cell 2012). Is the reduced viability reported by the authors specifically observed during differentiation in 3D cultures, or also in conventional 2i/LIF cultures?

The reduced viability of Esrrb knock out ES cells was reported also in conventional (2D) culture conditions. As mentioned above, the co-authors Kenjiro Adachi, Guangming Wu and Hans R. Schöler showed that Esrrb is required for ES cells maintenance in both Lif or ground state (2i/Lif) culture conditions (Adachi ... Schöler, Cell Stem Cell, 2018). In addition, Atlasi ...Stunnenberg, Nature Cell Biology, 2019 stated that "*We found that Esrrb^{-/-} ESCs lose self-renewal and proliferation capacity*" and Esrrb knock out "*compromised ESC survival and proliferation*".

[Redacted]

(Figure 6A from Atlasi ...Stunnenberg, Nature Cell Biology, 2019)

6/

The authors could consider discussing the potential role of Esrrb in regulating rosette formation and cavitation in the extraembryonic ectoderm (Christodoulou,..., Zernicka-Goetz; Nature Cell Biology 2018), where Esrrb remains expressed after implantation.

We grateful for Reviewer 1 suggestion. Although extraembryonic ectoderm (ExE) is a polarized tissue, the mechanism of trophoblast morphogenesis is fundamentally different compared to the epiblast. Very briefly, the apical-basal polarity in the epiblast is established de novo, whereas ExE (as Zernicka-Goetz lab proposed) is formed by folding the polar trophectoderm, which is already polarized.

Moreover, *Esrrb* is a Fgf/Erk, (not Wnt) target gene in the ExE, as previously shown by Latos ... Hemberger, Nature Comm 2015. Therefore, we consider that introducing and further comparing epiblast and trophoblast morphogenesis in the Discussion can result in loss of focus and even confusion for the reader. Such comparison will be suitable for a review article, where all the necessary background information can be gradually presented.

Reviewer #2 (Remarks to the Author):

The authors in this study investigated the signaling pathway to sustain the naïve pluripotency of epiblast cells. They show that active Wnt/ β -catenin signaling is critical to suppress the differentiation/epithelialization of embryonic stem cells (ES) in a 3D in vitro culture model. *Esrrb*, a downstream target of β -catenin, mediates the suppression of epithelialization as long as ES cells are in naïve pluripotent status. Although the in vivo roles of β -catenin and *Esrrb* during epiblast epithelialization is limited in normal pregnancy, they found that Wnt/ β -catenin signaling is active in epiblast of dormant embryos during diapause. Using Wntless null embryos, which are deficient in secretion of ligands, the authors suggest that Wnt ligands were produced by diapausing embryos but not the uterus.

We thank Reviewer 2 for his/her positive comments and suggestions that aided refining the manuscript. We have addressed all his/her comments in full below.

The mechanism of epiblast differentiation and loss of naïve pluripotency is not clearly understood. The investigators used genetic ES cell models and 3D in vitro culture system to mimic epiblast epithelialization. The identification of Wnt/ β -catenin signaling in suppressing epiblast differentiation is important. The manuscript is well organized and written.

Major concern is about the in vivo role of the Wnt/ β -catenin signaling in epiblast epithelization. The data suggest that the Wnt/ β -catenin signaling has a limited role in epiblast epithelization during normal pregnancy. Although Wnt/ β -catenin signaling is active in epiblast during diapause, the status of epiblast cells in diapause is different from that in the in vitro cultures and normal pregnancy. After the epithelization of epiblast cells in dormant embryos, the epiblast cells still maintain naïve pluripotency (Nanog positive, Figure 5b). However, in culture, ES cells lose Nanog while undergoing epithelization (Figure 1d). These results raise the question whether epiblast naïve pluripotency and its epithelization are mutually exclusive. Does the Wnt/ β -catenin signaling regulated cell polarity determine cell pluripotency in diapause embryos? Given the length of diapause which could be months in specific species, it may be questioned whether the embryo quality affects epithelization in the epiblast.

Our work revealed that diapause is not a static state, but a dynamic process. Indeed, the epiblast is Nanog positive during diapause, however we showed that Nanog expression does not control the establishment of epithelial polarity (Figures 3i and 3h). The process of epithelialization is controlled by *Esrrb*, which functions downstream of Wnt/ β -catenin signaling (Figures 3i and 3h).

Following Reviewer 2 query (in conjunction with the first question of Reviewer 1) we examined further the relation between Wnt/ β -catenin/*Esrrb* cascade and the process of epithelialization. We already showed that Wnt signaling activity peaks at EDG7.5 and decreases at EDG9.5, which is associated with the establishment of epithelial polarity at EDG9.5 (Figures 5c and 5d). We found that *Esrrb* expression is also high at EDG7.5 and decreases at EDG9.5 (new Figures S7c and S7d), indicating that *Esrrb* levels correlate with Wnt signaling activity. Accordingly, *Esrrb* expression was diminished in Wls knockout embryos at EDGE7.5 (new Figures 8c and 8d). Wls depletion also resulted in earlier epithelialization of the pluripotent lineage at EDG7.5 (Figure 8e).

Next, we sustained Wnt activity in the epiblast expressing β -catenin stabilized form to examine whether epithelialization will be affected at EDG9.5. Briefly, we aggregated β -catenin exon-3 deleted

and control exon-3 floxed ES cells with host morulae expressing Histone-H2B:GFP reporter allele. The resulting chimeric blastocysts comprised of wild-type (EGFP+) extraembryonic tissues and EGFP-epiblast comprised of the donor cells. These embryos were transferred into foster mother, where diapause was induced via ovariectomy and subsequently recovered at EDG9.5 (new Figures 7a and 7b). We found that embryos where β -catenin was stabilized displayed higher levels of Esrrb expression and low rate of epiblast polarization (new Figures 7b – 7d), compared to control embryos. This shows that active canonical Wnt/b-cat signaling maintains Esrrb expression during diapause and counters epithelialization of the pluripotent lineage.

Moreover, sustaining Esrrb expression alone in the epiblast using the similar experimental approach resulted in block of polarity at EDG9.5 (new Figures 7e and 7f). We aggregated ES cells that constitutively express Esrrb transgene or control wild-type ES cells with host morulae (EGFP+) to generate chimeric blastocysts. The embryos were transferred into foster mothers, where diapause was induced via ovariectomy, and subsequently recovered at EDG9.5. Control embryos exhibited epiblast polarization, whereas the pluripotent lineage of embryos ectopically expressing Esrrb remained non-polarized (new Figures 7e and 7f). Taken together, this new data shows that the Wnt/b-cat/Esrrb signaling controls the epithelialization of the epiblast.

Concerning embryo quality, we are extremely careful and we have a lot of experience with these developmental stages. In our analysis, we always compare experimental (e.g. knock out) to control (e.g. heterozygous and wild type) littermates and the effects that we report are because of the specific genetic alterations. Moreover, we are not the first to induce diapause, there are multiple studies where diapause was induced and then embryos were reactivated at various time points (EDG6.5, EDG7.5, EDG8.5, EDG9.5 or EDG 11.5) and these embryos implanted and continued to develop normally (Susan MacLean Hunter and Martin Evans, *Mol Rep and Dev*, 1998; Hussein... Ruohola-Baker, *Dev Cell* 2020; Bulit-Karslioglu...Ramalho-Santos, *Nature* 2016; Nichols and Smith 2001). Diapause was sustained experimentally even for 30-40 days and embryos were still recovered (Weitlauf & Greenwald, 1968, Weitlauf 1971). In our experiments we never induce diapause for such extremely long time, we always analyse embryos within the first week of diapause, further ensuring that embryo quality is not compromised.

In line 63, the authors stated that “In mice, by embryonic day four and a half (E4.5), the specification of these three lineages is completed, establishing a mature blastocyst. After the E4.5 embryo hatches from its glycoprotein envelope (Zona pellucida), it comes into direct contact with the mother and initiates implantation into the uterine wall.” The pregnancy timing described in the paragraph is not correct. Blastocysts hatch on E3.5, and implant in the midnight of day 4 of pregnancy. Day 1 is considered finding a vaginal plug in the morning. The reference cited for this time-line should be replaced by more appropriate citation.

We follow a standard timeline of developmental stages – zygote (E0.5), 2-cell stage (E1.5), morula (E2.5), early blastocyst (E3.5) and late blastocyst (E4.5), that can be found in textbooks and manuals such as “Manipulating the mouse embryo” and multiple research papers and reviews, such as “Blastocyst lineage formation, early embryonic asymmetries and axis patterning in the mouse” Janet Rossant, Patrick P. L. Tam., *Development* 2009

[Redacted]

The authors stated that “a temporary spike in the levels of ovarian estrogen at day 4 of pregnancy leads to the removal of this protective layer, establishing a limited receptive phase”. Embryos downregulate maternal MUC1 expression. However, preimplantation estrogen is not an absolute requirement for zona dissolution. They suggest *Esrrb* is downstream of β -catenin, but did not provide direct evidence. In figure 3h, an experiment of overexpression of *Esrrb* in β -catenin deleted ES could prove *Esrrb* is downstream of β -catenin.

Reviewer 2 is correct; estrogen is not an absolute requirement for hatching and we never claimed this. The ovarian estrogen is required for the establishment of the so-called implantation window, a limited time of uterine receptivity (reviewed in Wang, H. & Dey, S. K. Roadmap to embryo implantation: clues from mouse models. *Nat Rev Genet*). The protective layer that we talk about is not Zona pellucida of the embryo, but the mucin layer covering the luminal epithelium of the uterus.

Esrrb is a well-established Wnt signaling target in ES cells (for example “*Esrrb* Is a Pivotal Target of the Gsk3/Tcf3 Axis Regulating Embryonic Stem Cell Self-Renewal” Martlo... A. Smith, *Cell Stem Cell* 2012). In agreement with this study we showed also find that 1) *Esrrb* expression changes upon CH treatment and 2) *Esrrb* is bound by Tcf3 and also changes upon deletion of Tcf3 (using available datasets of Marson, A. et al. *Cell* 2008 and Yi, F. et al. *Nat Cell Biol* 2011 (Figures 3e and 3f , new Figure S2e, new Table S3). Following Reviewer 2 request, we examined this further. We found that in vitro (in 3D ES cell culture) stabilizing β -catenin following exon-3 deletion resulted in sustained *Esrrb* expression, in comparison to control exon-3 floxed ES cells (new Figure S3d). The same experiments were also done in vivo, as mentioned above (new Figures 7a – 7d).

Reviewer 2 also suggested that “*experiment of overexpression of Esrrb in β -catenin deleted ES could prove Esrrb is downstream of β -catenin*”. As β -catenin knock out ES cells show an adhesion defect (Figures 2j-2i), we used the E-cadherin-a-catenin fusion to compensate for the adhesion defect in β -catenin null cells and overexpressed *Esrrb* in both DMSO and 2i culture conditions. Ectopic expression of *Esrrb* resulted in block of polarity in the absence β -catenin (new Figure S3e, see also Figures 2g and 2j), indicating the *Esrrb* is downstream of β -catenin.

In figure S2f, how could the authors differentiate trophoblast cells and epiblast cells?

As we added new data, this figure is now labelled as S3f. To discriminate the trophoblast cells, we provided new images of embryos stained for Eomes (trophoblast marker), in combination with Esrrb and Sox17 (new Figure S3f).

In figure 4g and 4j, given Ntn1 and Spry2 decrease in Esrrb null, the deletion of Spry2 in ES cells could show more phenotype? Figure 4i showed some effect of Spry2 overexpression on cell polarization, suggesting that Spry2 may play a role during the process. Whether Esrrb controls the epithelial program via Spry2 is still an open question.

Spry2 ko mice are born and exhibit hearing defects (Shim et al, Dev Cell 2005) and combined deletion of Spr2 and Spry4 show embryonic defects and die around E12.5 (Taniguchi et al, BBRC, 2007). As the Spry family contains four members, Spry1, Spry2, Spry3, and Spry4, ablation of the whole family may result in even earlier phenotype.

It is very important to clarify that Spry2 is most likely one factor a whole group that modulates the epithelial program. We stated in the Result section that *“Still, a large portion of the ES cell clumps established apical-basal polarity, suggesting that additional factors are involved in regulating the epithelial program downstream of Esrrb.”* In addition, we underlined this in the Discussion by stating that *“Spry2 may belong to a larger, yet unidentified network of effector proteins involved in both modulating cell fate specification and tissue morphology of the pluripotent lineage.”* To make sure that this message is clear for the reader we removed Spry2 from the final Figure 8g and updated the model.

Reference 33 used embryos from EDG6.5 but not 7.5.

We are grateful to Reviewer 2 that noticed this and we corrected the main text and the figures.

In figure 5e, the quality of the dormant embryo on EDG5.5 is not of high caliber.

We updated the figure.

Just a note, the blastocoel cavity dynamically expands and collapses and some embryos can appear with smaller cavity at the time of isolation and fixation.

The model of Wntless is not a clean model, since the Wntless null embryos are not able to release any ligands, not limited to Wnt ligands. The interaction between embryos and uteri is mutual. The model could stop the secretion from embryos to uteri, and indirectly interferes the maternal secretion.

Wntless loss of function is a very well established model system to study Wnt signalling. Banzinger et al Cell, 2006 in their paper *“Wntless, a Conserved Membrane Protein Dedicated to the Secretion of Wnt Proteins from Signaling Cells”* showed that other pathways such as Dpp, EGF, Notch, and Insulin are not dependent on Wntless. There is no evidence that mouse embryos deficient for Wls are not able to release any ligands at all. For example, if only Fgf signaling is perturbed, the embryos will not form primitive endoderm and obviously this is not the case with Wls mutants. Wls knock out

embryos phenocopy Wnt3 ko (Fu... Hsu, PNAS 2009). Moreover, conditional deletion of Wntless at different developmental stages also phenocopy previously characterized Wnt mutants (Carpenter ... Lang, Genesis 2010).

Since diapause occurs within the uterus, not in in vitro settings, epiblast epithelialization must involve factors from the uterus and/or withdrawal of some signals emanating from the uterus. An in vitro model to induce blastocyst diapause could answer many of the concerns.

We completely agree with Reviewer 2 comment that diapause emerges in in vivo settings where the embryo and the mother communicate. Whether the process of epithelialization during diapause requires maternal input or is an embryo intrinsic process, is an interesting and still an open question. Future establishment of an in vitro model mimicking the maternal environment will be extremely beneficial for the developmental biology community and will be essential to understand epiblast morphogenesis. Such system, potentially combined also with the recently established “artificial embryos / blastoids”, will be a very powerful tool to model and study early embryogenesis.

Reviewers' Comments:

Reviewer #1:

Remarks to the Author:

In this revised version, the manuscript from Bedzhov and collaborators has substantially improved. The authors now presents clear evidence that WNT signaling counters the acquisition of epithelial character in naïve pluripotent cells at least in part by sustaining Esrrb expression, and present data supporting this conclusion both in the embryo and in culture. Further, the gene expression analyses underlying the choice of the candidates to be functionally tested as downstream mediators of WNT and Esrrb have been sufficiently extended and are more clearly presented. Overall, the authors have satisfactorily address all previous concerns.

Reviewer #2:

Remarks to the Author:

The authors' comments are largely acceptable and the manuscript is now recommended for acceptance, except one clarification of inducing diapause with Tamoxifen. It was shown that Tamoxifen mediates estrogenic effects in the mouse uterus (Role of early and late oestrogenic effects on implantation in the mouse. *J Reprod Fertil* 81(2):453 458, 1987). The authors should clarify this point.

Response to reviewers' comments:

Reviewer #1 (Remarks to the Author):

In this revised version, the manuscript from Bedzhov and collaborators has substantially improved. The authors now presents clear evidence that WNT signaling counters the acquisition of epithelial character in naïve pluripotent cells at least in part by sustaining Esrrb expression, and present data supporting this conclusion both in the embryo and in culture. Further, the gene expression analyses underlying the choice of the candidates to be functionally tested as downstream mediators of WNT and Esrrb have been sufficiently extended and are more clearly presented. Overall, the authors have satisfactorily address all previous concerns.

We are grateful to Reviewer 1 for his/her constructive suggestions that helped to improve the manuscript.

Reviewer #2 (Remarks to the Author):

The authors' comments are largely acceptable and the manuscript is now recommended for acceptance, except one clarification of inducing diapause with Tamoxifen. It was shown that Tamoxifen mediates estrogenic effects in the mouse uterus (Role of early and late oestrogenic effects on implantation in the mouse. J Reprod Fertil 81(2):453 458, 1987). The authors should clarify this point.

We thank Reviewer 2 for his/her time and suggestions. The induction of diapause, as a result of tamoxifen administration, is well-established approach that was published first by Martin Evans (Evans and Fawcett 1983) and used in subsequent studies (Hirata et al 2002). Details

Hunter, S.M., and Evans, M. (1999). Non-surgical method for the induction of delayed implantation and recovery of viable blastocysts in rats and mice by the use of tamoxifen and Depo-Provera. *Mol Reprod Dev* 52, 29-32.

Nichols, J., and Ying, Q.L. (2006). Derivation and propagation of embryonic stem cells in serum- and feeder-free culture. *Methods Mol Biol* 329, 91-98.